# Predictable Compression Failures:
# Order Sensitivity and Information Budgeting for Evidence-Grounded Binary Adjudication

**Leon Chlon** [1]   **Ahmed Karim** [2]   **MarcAntonio Awada** [3]

## Abstract

Transformers used for evidence-grounded binary adjudication (e.g., support/refute, yes/no, or verifier-backed pass/fail decisions) can be sensitive to the order in which exchangeable evidence is presented, producing dispersion across permutations and unreliable attempted answers under a verifier-relative Bernoulli predicate. We treat evidence order as a nuisance variable and formalize an expectation–realization gap: next-token training can minimize expected conditional description length over orderings while a fixed ordering remains position-sensitive. Our Quantified Martingale Violation (QMV) bound predicts the dispersion induced by adjacent-rank positional sensitivity, with $O(\log n)$ growth in the harmonic regime; our Expectation-level Decompression Law (EDFL) specializes a KL convexity/data-processing bound to Bernoulli predicates, yielding Bits-to-Trust (B2T), Risk-of-Hallucination (RoH), and an Information Sufficiency Ratio (ISR) gate for answer/abstain decisions. On 3,059 grounded items from FEVER, HotpotQA, NQ-Open, PopQA, and Controls, we observe logarithmic dispersion and positive Jensen gains from uniform permutation mixtures. In one pre-specified held-out audit (528 items), the analytically fixed ISR$= 1$ gate attains 0.0–0.7% hallucination with 20.6–27.9% abstention (95% CIs), supporting the operating point without claiming universal calibration across all model families or unrestricted generation.

## 1. Introduction

Transformers with positional encodings can be strong in-context learners, yet their predictions can vary substantially under *permutations of the same evidence* (Falck et al., 2024; Liu et al., 2024). This is a reliability problem when the evidence set is intended to be exchangeable: reordering retrieved chunks should not change whether the evidence supports an answer, but in practice it can change the model's probability of answering correctly or abstaining.

We study this problem for evidence-grounded *Bernoulli adjudication*. The output may be a support/refute label, a yes/no decision, a multiple-choice correctness predicate, a discrete-tool success predicate, or a post-hoc verifier result. The common object is a predicate $g : \mathcal{Y} \to \{0, 1\}$ with a reference distribution $P_{\text{ref}}$ induced by a specification, a deterministic tool, gold labels in evaluation, or a calibrated verifier. The guarantees are always relative to that predicate and reference; unrestricted generation without a verifier is outside the scope of this paper.

The reader's map is simple. For each evidence ordering $\pi$, the model produces a Bernoulli probability $q_\pi = S_\pi(g = 1)$. We average these probabilities over permutations to obtain $\bar{q}$, track the conservative lower value $q_{\text{lo}}$, and estimate an information budget $\bar{\Delta}$ from Bernoulli KL terms. QMV explains when order-induced dispersion should grow with context depth. EDFL converts a target reliability $p^\star = 1 - h^\star$ into the minimum budget needed to trust an answer. The Information Sufficiency Ratio, ISR $= \bar{\Delta}/\text{B2T}$, then gives a fixed rule: answer when ISR $\geq 1$, otherwise abstain or acquire more evidence.

This framing reconciles two facts. Next-token training minimizes cross-entropy on sequences, which corresponds to minimizing *expected* conditional description length over orderings; a model can therefore be close to Bayes-optimal on average while deviating from a permutation-invariant predictor at any fixed ordering. Our contribution is to quantify this expectation–realization gap and turn it into a deployment-facing answer/abstain interface.

---

[1]University of Oxford [2]University College London [3]Division of Continuing Education, Harvard University. Correspondence to: Leon Chlon <lc574@cantab.ac.uk>.

*Proceedings of the 43$^{rd}$ International Conference on Machine Learning*, Seoul, South Korea. PMLR 306, 2026. Copyright 2026 by the author(s).

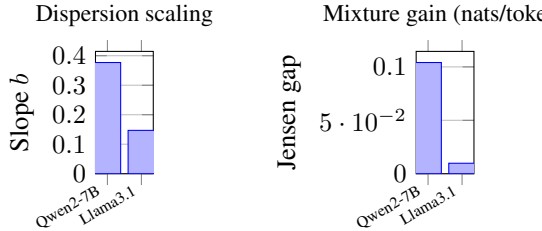

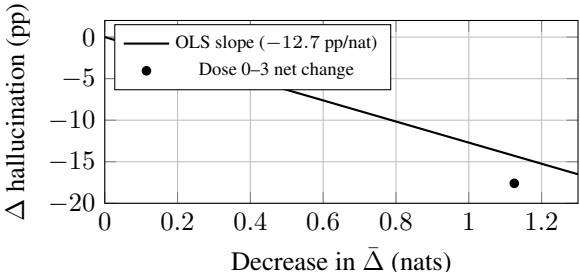

*Figure 1.* **Order sensitivity and mixture gains (Experiment 1;** $N = 3{,}059$, $m = 16$ **banded permutations).** Left: estimated log-scaling slope $b$ in the empirical law $|R_\pi| \approx a + b \log n$ (95% CI as defined in Section 4.1; Table 3). Right: Jensen gap (nats/token) from uniform permutation mixtures, quantifying the cross-entropy improvement from averaging over evidence orderings.

## 1.1. Our Contributions

We make three contributions:

- **Quantified order sensitivity.** We prove a *Quantified Martingale Violation* (QMV) bound controlling permutation-induced dispersion, with explicit constants and $O(\log n)$ scaling under harmonic positional sensitivity (Theorem 1 and Theorem 2).

- **EDFL as an actionable Bernoulli specialization.** EDFL (Theorem 5) is a Bernoulli coarse-graining of a convexity + data-processing bound, yielding closed-form planners (B2T/RoH/ISR) and a fixed ISR=1 answer/abstain rule; we also derive a rare-event lower bound scaling like $\log(1/\bar{q})$.

- **A deployable answer/abstain rule.** From EDFL we define Bits-to-Trust (B2T), Risk-of-Hallucination (RoH), and the Information Sufficiency Ratio (ISR), and give an ISR-gating procedure under permutation mixtures (Algorithm 1).

## 1.2. Empirical Validation

We validate the theory using observables not defined by the planners themselves (Appendix B). In particular:

- **Permutation mixtures improve log-loss and accuracy** (Experiment 1), and dispersion follows $a + b \log n$ across depths (Figure 1).

- **Randomized dose-response identifies a causal effect of support dose** on the information-budget estimate and hallucination, holding prompt length fixed (Experiment 2; Figure 2).

- **A pre-specified held-out audit** reports an out-of-sample operating point for ISR gating on the reliability–coverage plane without post-hoc threshold tuning (Experiment 3; Figure 3).

*Figure 2.* **Causal dose-response of support dose (Experiment 2).** Randomized study holding prompt length fixed at $L{=}4$ chunks while varying support dose $d \in \{0, 1, 2, 3\}$ (subset of Factuality Slice; Appendix H.4). We plot the net change from dose $0 \to 3$ in hallucination rate against the induced *decrease* in the information-budget estimate ($\Delta\bar{\Delta} := \bar{\Delta}_{d=0} - \bar{\Delta}_{d=3} \approx 3 \times 0.375 = 1.125$ nats). The fitted slope (OLS) is 0.127 fewer hallucinations per additional nat ($-12.7$ pp/nat); over the same shift, answer rate increases by 37.5pp and accuracy on attempts by 45.6pp. The causal intervention is support dose; the information-budget interpretation follows from the measured first stage rather than being the randomized treatment itself.

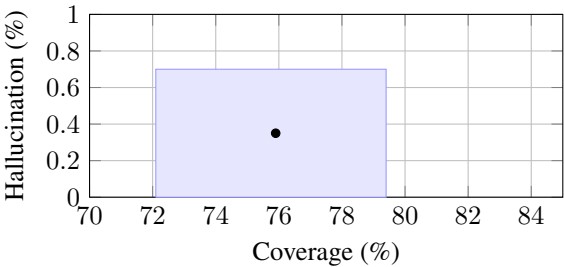

*Figure 3.* **Audit operating point (Experiment 3;** $N = 528$, $m = 6$**).** Pre-specified held-out audit with ISR$= 1$ and fixed seeds. Shaded region shows the 95% CI for coverage (1–abstention) and hallucination rate (Wilson intervals as in Section 4.1); midpoint shown for visualization.

## 2. Related Work

**Order sensitivity and positional structure.** Transformers can be highly sensitive to evidence order, including systematic "lost-in-the-middle" effects (Liu et al., 2024), and martingale-based tests show that LLM in-context learning can violate exchangeability requirements expected of Bayesian learning systems (Falck et al., 2024). We quantify the resulting dispersion under an explicit positional-sensitivity model and pair it with an assumption-free Jensen–Shannon certificate. RoPE-style positional mechanisms provide one motivation for studying decaying positional influence: RoFormer introduced relative-distance decay as a useful property of rotary embeddings (Su et al., 2024), later analyses connect RoPE base choices to long-context behaviour (Xu et al., 2024) and position-sensitive query/key structure (Chen & Yan, 2024), while Barbero et al. (Barbero et al., 2025) caution that decay-only explanations are

incomplete.

**Bayesian/MDL views of in-context learning.** Prior work connects transformers and in-context learning to Bayesian inference in various forms (Xie et al., 2022; Zhang et al., 2024; Bai et al., 2023; Reuter et al., 2025). The compression–learning connection is classically captured by Minimum Description Length (MDL) (Grünwald, 2007), and language modeling can be studied directly through a compression lens (Delétang et al., 2024). Our contribution is to make the expectation-vs.-realization gap over evidence orderings explicit and measurable, rather than treating order invariance as an implicit property of the predictor.

**Hallucination control, calibration, and abstention.** There is extensive work on detection and calibration signals, including semantic entropy for detecting confabulations (Farquhar et al., 2024) and limits showing that calibrated language models can still hallucinate under appropriate definitions (Kalai & Vempala, 2024). Our setting is narrower: evidence-grounded Bernoulli adjudication with a verifier-relative predicate. Within that setting, EDFL yields closed-form B2T/RoH/ISR planners and a fixed ISR= 1 answer/abstain rule.

**Perturbation ensembling and selective prediction.** Self-consistency and related ensemble methods aggregate over sampled reasoning paths or other perturbations to improve answer accuracy and stability (Wang et al., 2023; Xu et al., 2025; Chang et al., 2024). Permutation mixtures answer a different question: they marginalize over orderings of the same exchangeable evidence multiset to estimate evidence sufficiency under a nuisance variable. A generic ensemble can be stably wrong when the evidence is inadequate; ISR is designed to reject such cases through a fixed budget rule. This interface is related to selective classification and reject-option risk–coverage tradeoffs (Chow, 1970; Geifman & El-Yaniv, 2017), but the selector is derived from an information budget rather than fit as a held-out confidence threshold.

## 3. Theory: Order-Sensitive Information-Theoretic Guarantees

We formalize how positional processing can create systematic deviations from permutation-invariant behaviour while preserving average-case optimality over orderings. The results below are scoped to evidence sets whose order is intended to be a nuisance; tasks where sequence order carries the semantics of the problem are outside the exchangeable-evidence regime.

### 3.1. Setup and Notation

Let $x$ denote an instance consisting of $n$ evidence chunks $X = (x_1, \ldots, x_n)$ and a target variable $Y$. Let $\pi$ be a permutation of chunk indices $\{1, \ldots, n\}$ and let $\Gamma_\pi$ reorder the chunks, $\Gamma_\pi(X) = (x_{\pi(1)}, \ldots, x_{\pi(n)})$. Let $S_\pi(\cdot)$ denote the model's predictive distribution over $Y$ given $\Gamma_\pi(X)$. We use $P$ to denote a target distribution over $Y$ (ground truth in supervised evaluation). In deployment, we instantiate $P$ with a reference distribution $P_{\text{ref}}$ induced by the application's specification for a predicate $g$ (gold labels in supervised evaluation; deterministic verification primitives such as database lookups, unit tests, or rules; human review; or a calibrated verifier). Our guarantees are stated relative to $P_{\text{ref}}$.

Importantly, we do *not* need to construct a reference distribution over free-form outputs. For a binary adjudication predicate $g : \mathcal{Y} \to \{0, 1\}$, define the induced (pushforward) distributions on $\{0, 1\}$:

$$P_{\text{ref},g}(z) := P_{\text{ref}}(g(Y) = z),$$
$$S_{\pi,g}(z) := S_\pi(g(Y) = z), \qquad z \in \{0, 1\}.$$

By data processing (coarse-graining cannot increase KL), $\text{KL}(P_{\text{ref}} \| S_\pi) \geq \text{KL}(P_{\text{ref},g} \| S_{\pi,g})$, so the relevant reference object for answer/abstain decisions is the Bernoulli parameter $p_{\text{ref}} := P_{\text{ref},g}(1)$. For a Bernoulli predicate $g : \mathcal{Y} \to \{0, 1\}$ (e.g., correctness, answer-vs-refuse, constraint satisfaction), define $q_\pi(x) := S_\pi(g=1)$ and $\bar{q}(x) := \mathbb{E}_\pi[q_\pi(x)]$. We assume $P_{\text{ref}} \ll S_\pi$ for all $\pi$ (absolute continuity), smoothing zero-probability tokens with $\varepsilon = 10^{-9}$ in practice. We write $E_{\text{pair}} := \mathbb{E}_{\pi,\pi'}|q_\pi(x) - q_{\pi'}(x)|$ for the expected pairwise absolute difference across independent permutations (with $\pi, \pi'$ i.i.d. uniform over permutations). All expectations over $\pi$ are conditional on fixed instance $x$.

All information quantities (KL divergences, information budgets, Jensen gaps) are reported in nats unless otherwise noted. The operational quantities used later are $q_{\text{lo}}(x) := \min_k q_{\pi_k}(x)$, $\bar{\Delta}(x) := m^{-1} \sum_k \text{KL}(\text{Ber}(p_{\text{ref}}) \| \text{Ber}(q_{\pi_k}))$ after the stated clipping convention, $\text{B2T}(x; p^\star) := \text{KL}(\text{Ber}(p^\star) \| \text{Ber}(q_{\text{lo}}(x)))$, and $\text{ISR}(x) := \bar{\Delta}(x) / \text{B2T}(x; p^\star)$.

### 3.2. Quantified Martingale Violations

**How to read QMV.** QMV is a conditional structural theorem, not a universal claim that every deployed transformer has harmonic positional sensitivity. It states that if the Bernoulli logit has local adjacent-rank sensitivity with a given decay profile, then permutation-induced dispersion follows the corresponding partial-sum law: polynomial for $\alpha < 1$, logarithmic for $\alpha = 1$, and bounded for $\alpha > 1$. The practical case for permutation mixtures does not depend on this assumption alone; Proposition 2 gives an assumption-free Jensen–Shannon certificate.

**Definition 1** (Permutation-induced residual). *For a Bernoulli predicate $g$ and position-aware predictions $q_\pi(x) := S_\pi(g=1)$, the permutation residual is $R_\pi(x) :=$*

$q_\pi(x) - \bar{q}(x)$ *where* $\bar{q}(x) := \mathbb{E}_\pi[q_\pi(x)]$.

**Assumption 1** (Local rank stability with bounded total variation). *For a fixed item $x$, define $f_x(\pi) := \text{logit}(q_\pi(x))$ on $S_n$. For each chunk $i \in \{1, \ldots, n\}$ and rank $t \in \{1, \ldots, n-1\}$ define the coordinate-wise adjacent increment*

$$\Delta_{i,t} := \sup_{\pi_{-i}} \left| f_x\big(r_i{=}t{+}1, \pi_{-i}\big) - f_x\big(r_i{=}t, \pi_{-i}\big) \right|,$$

*where $\pi_{-i}$ ranges over permutations of the remaining chunks and $r_i$ is the rank of chunk $i$. Let the coordinate total variation be $\text{TV}_i(x) := \sum_{t=1}^{n-1} \Delta_{i,t}$ and suppose*

$$\sum_{i=1}^n \text{TV}_i(x) \leq B(x) < \infty.$$

*Moreover, if there exist nonnegative coefficients $C_i$ and $\alpha > 0$ such that $\Delta_{i,t} \leq C_i t^{-\alpha}$ and $\sum_i C_i =: C_{\text{tot}}(x) < \infty$, we say the decay is $(\alpha, C_{\text{tot}})$-regular.*

**Theorem 1** (Quantified Martingale Violation). *Under Assumption 1,*

$$\mathbb{E}_\pi \big| R_\pi(x) \big| \leq \mathbb{E}_{\pi,\pi'} \big| q_\pi(x) - q_{\pi'}(x) \big| \leq \frac{1}{4} \sum_{i=1}^n \text{TV}_i(x).$$

*If, in addition, the decay is $(\alpha, C_{\text{tot}})$-regular, then*

$$\mathbb{E}_\pi \big| R_\pi(x) \big| \leq \frac{C_{\text{tot}}(x)}{4} \times$$
$$\begin{cases} \frac{1}{1-\alpha}\big(n^{1-\alpha} - 1\big), & \alpha \in (0,1), \\ H_{n-1} = \log n - \gamma + o(1), & \alpha = 1, \\ \zeta(\alpha) + o(1), & \alpha > 1. \end{cases}$$

**In words.** QMV bounds permutation-induced dispersion by the cumulative sensitivity of the logit to adjacent rank changes. Under harmonic decay ($\alpha = 1$), the bound grows like $\log n$, motivating the empirical $a + b \log n$ scaling test in Experiment 1. Failure of the log law would not invalidate the gate; it would instead indicate a different positional regime or a chunking/permutation scheme outside the modeled assumptions.

**Assumption 2** (First-order positional sensitivity). *There exist nonnegative content weights $w_1, \ldots, w_n$ with $\sum_i w_i = 1$ and a potential $\psi$ that is $(\alpha, C)$-regular (i.e., $|\psi(r + 1) - \psi(r)| \leq Cr^{-\alpha}$) such that:*

$$\text{logit}(q_\pi(x)) = a(x) + \sum_{i=1}^n w_i \psi(pos_\pi(i))$$

**Proposition 1** (First-order model $\Rightarrow$ local stability). *Suppose Assumption 2 holds with nonnegative weights $w_i$ summing to 1 and $\psi$ $(\alpha, C)$-regular. Then Assumption 1 holds with $\Delta_{i,t} \leq w_i C t^{-\alpha}$ and hence $C_{\text{tot}}(x) = C$.*

**Theorem 2** (Explicit log-scaling under first-order positional sensitivity). *Under Assumption 2 with harmonic decay ($\alpha = 1$), the permutation-induced dispersion admits an explicit $O(\log n)$ upper bound with constants:*

$$\mathbb{E}_\pi |R_\pi(x)| \leq \frac{C}{4}\Big(\log n - \frac{3}{2} + o(1)\Big)$$

*More generally: $\alpha < 1 \Rightarrow O(n^{1-\alpha})$; $\alpha = 1 \Rightarrow O(\log n)$; $\alpha > 1 \Rightarrow O(1)$ due to harmonic versus p-series convergence.*

**Proposition 2** (Assumption-free JS certificate). *Let $\bar{S} = \mathbb{E}_\pi S_\pi$. For any Bernoulli predicate $g$ with $q_\pi = S_\pi(g{=}1)$ and $\bar{q} = \bar{S}(g{=}1)$,*

$$\mathbb{E}_\pi \big| q_\pi - \bar{q} \big| \leq \mathbb{E}_\pi \text{TV}(S_\pi, \bar{S}) \leq \sqrt{\frac{1}{2} \mathbb{E}_\pi \text{KL}(S_\pi \| \bar{S})}.$$

*Note $\mathbb{E}_\pi \text{KL}(S_\pi \| \bar{S})$ is the (generalized) Jensen-Shannon divergence (JSD) with uniform weights, hence the bound controls dispersion by JSD via Pinsker.*

### 3.3. MDL Optimality Through Architectural Closure (conceptual)

**How to read this subsection.** The results below are *representational*: they formalize that finite permutation mixtures are realizable if one allows tied-weight multi-branch evaluation with an averaging head (i.e., ensembling "inside" the hypothesis class). This is a conceptual lens on expectation-versus-realization gaps and does *not* claim that a standard single-branch deployed transformer literally implements the multi-branch architecture or that SGD recovers the mixture without additional structure. In practice, expectation over orderings comes either from training-time exposure to randomized orderings or from inference-time permutation mixtures.

**Theorem 3** (Permutation-mixture realizability with averaging head). *Fix any base model $p_\theta(y|x)$ in the model family and any finite set of permutations $\Pi$. Consider an ensemble-within-the-network that:*

(i) *Applies the same parameters $\theta$ to $x$ along each branch after permuting inputs by $\Gamma_\pi$ for every $\pi \in \Pi$*

(ii) *Outputs per-branch distributions $p_\theta(y|\Gamma_\pi(x))$*

(iii) *Averages the distributions in probability space with equal weights*

*Then the composite network implements:*

$$q_{\theta, \Pi}(y|x) = \frac{1}{|\Pi|} \sum_{\pi \in \Pi} p_\theta(y|\Gamma_\pi(x))$$

*If the model class includes such tied-weight multi-branch compositions with a linear averaging head, it is closed under finite permutation mixtures. As $|\Pi| \to \infty$ along*

*i.i.d. draws from the uniform measure over permutations, $q_{\theta,\Pi} \Rightarrow \bar{p}_\theta(y|x) = \mathbb{E}_\pi[p_\theta(y|\Gamma_\pi(x))]$ almost surely.*

*Proof.* The only operation beyond the base model is a convex combination in probability space. Since each branch outputs a normalized distribution, the average is also a normalized distribution and is implementable by a fixed linear layer that sums corresponding probabilities across branches with weights $1/|\Pi|$. Tied weights ensure the branches are copies of $p_\theta$ evaluated at permuted inputs, so the composite realizes the desired mixture. By the strong law of large numbers, the empirical average over i.i.d. permutations converges pointwise to the permutation expectation. $\square$

**Theorem 4** (MDL Optimality in Expectation). *Under the architectural closure of Theorem 3, the permutation-mixture family can achieve information-theoretic optimality in expectation over orderings. Write $P_T := P(\cdot|T(X))$ and $p_{\theta,\pi} := p_\theta(\cdot|\Gamma_\pi(X))$. The risk decomposes as:*

$$\mathbb{E}_{X,Y,\pi}\big[-\log p_{\theta,\pi}(Y)\big] = H(Y|T)$$
$$+ \mathbb{E}_{X,\pi} \mathrm{KL}\big(P_T \| p_{\theta,\pi}\big)$$

*The gap between expectation and realization is captured by the positional Jensen penalty:*

$$\mathfrak{J}_\Gamma(P,\theta) := \mathbb{E}_\pi[\mathrm{KL}(P\|p_{\theta,\pi})] - \mathrm{KL}(P\|\bar{p}_\theta) \geq 0$$

*Over the convex hull of permutation mixtures realized by the architecture, the I-projection onto the exchangeable target $P(\cdot|T(X))$ yields:*

$$\inf_\theta \mathbb{E}_{(X,Y)}\mathbb{E}_\pi\big[-\log p_{\theta,\pi}(Y)\big] = H(Y|T) + o(1)$$

*Proof.* The risk decomposition follows from the chain rule for KL divergence. Since Theorem 3 guarantees the model family contains $\bar{p}_\theta$, the convex hull of permutation mixtures is realizable. The I-projection of $P(\cdot|T(X))$ onto this convex hull minimizes the KL divergence, and when the target is realizable (i.e., exchangeable and in the convex hull), the residual term vanishes, yielding MDL optimality in expectation. $\square$

### 3.4. The Expectation-level Decompression Law

Binary adjudication is the natural unit of analysis as EDFL provides closed-form bounds for Bernoulli events and production systems make binary answer/abstain decisions (we evaluate via Bernoulli predicates $g$, details in Appendix B).

**Theorem 5** (Expectation-level Decompression Law (EDFL)). *For any event $\mathcal{A}$ with prior mass $\bar{q} = \bar{S}(\mathcal{A})$ and posterior mass $p = P(\mathcal{A})$, the expected information budget satisfies:*

$$\bar{\Delta} := \mathbb{E}_\pi[\mathrm{KL}(P\|S_\pi)] \geq \mathrm{KL}(\mathrm{Ber}(p)\|\mathrm{Ber}(\bar{q}))$$

*with equality when $P$ is the I-projection of $\bar{S}$ onto $\{Q : Q(\mathcal{A}) = p\}$. Equality holds for the exponentially tilted distribution $P^\star(y) \propto \bar{S}(y)e^{\lambda g(y)}$ with $\lambda$ chosen so $P^\star(\mathcal{A}) = p$ (standard I-projection).*

*Remark* 1 (Bernoulli instantiation). Fix a predicate $g$ and consider the induced distributions $P_{\mathrm{ref},g}$ and $S_{\pi,g}$ on $\{0,1\}$. Applying EDFL on this coarse-grained space yields the information budget

$$\bar{\Delta}_g := \mathbb{E}_\pi[\mathrm{KL}(P_{\mathrm{ref},g}\|S_{\pi,g})]$$
$$= \mathbb{E}_\pi\big[\mathrm{KL}(\mathrm{Ber}(p_{\mathrm{ref}})\|\mathrm{Ber}(q_\pi))\big],$$

which depends only on Bernoulli probabilities. By data processing, $\mathbb{E}_\pi[\mathrm{KL}(P_{\mathrm{ref}}\|S_\pi)] \geq \bar{\Delta}_g$, so working in Bernoulli space is conservative when the downstream decision is binary.

**Role of EDFL.** EDFL is not introduced as a new inequality in isolation: its backbone is KL convexity plus data processing (Appendix A.4). The contribution is the Bernoulli operationalization for evidence-grounded adjudication: closed-form B2T/RoH/ISR planners and a fixed ISR$= 1$ answer/abstain boundary that can be audited out of sample.

**In words.** EDFL lower-bounds the expected information budget needed to shift an event's probability from prior mass $\bar{q}$ to posterior mass $p$. For rare events, the required budget grows like $\log(1/\bar{q})$ (Corollary 1), which motivates B2T as a minimum budget for target reliability and the ISR gate as a deployment-time rule.

**Corollary 1** (Compression Failure for Rare Events). *For fixed $p = 1 - \varepsilon$ with $\varepsilon \in (0, \frac{1}{2}]$, when $\bar{q} \ll 1$, achieving reliability $p = 1 - \varepsilon$ requires:*

$$\bar{\Delta} \geq (1-\varepsilon)\log\frac{1}{\bar{q}} + O(\bar{q})$$

*As a uniform lower bound for all $\varepsilon \in (0, \frac{1}{2}]$:*

$$\bar{\Delta} \geq \frac{1}{2}\log\frac{1}{\bar{q}} - \log 2 + O(\bar{q})$$

*Insufficient information can lead to failed attempted answers under the Bernoulli predicate.*

### 3.5. Operational Planners

EDFL uses $p$ to denote the (unknown) achieved success probability of an event under the reference distribution. In contrast, planning specifies a target success probability $p^\star = 1 - h^\star$, and B2T/ISR are computed relative to this target.

The ISR threshold is fixed analytically at 1.0; permutations, clipping ($B=6$), and seeds were fixed before scoring, making boundary alignment a falsifiable out-of-sample check.

*Table 1.* **Box 1: operational planners for Bernoulli adjudication.** The quantities are computed from permutation-mixture probabilities for a verifier-relative predicate $g$.

| Quantity | Meaning |
|---|---|
| $\mathrm{B2T}(x; p^\star)$ | Minimum budget needed to trust an answer at target reliability $p^\star = 1 - h^\star$: $\mathrm{KL}(\mathrm{Ber}(p^\star) \,\|\, \mathrm{Ber}(q_{\mathrm{lo}}(x)))$. |
| $\mathrm{RoH}(x)$ | Residual risk implied by the measured budget: $1 - p_{\max}(\bar{\Delta}(x), \bar{q}(x))$, where $p_{\max}$ solves the Bernoulli EDFL inequality. |
| $\mathrm{ISR}(x)$ | Available budget divided by required budget: $\bar{\Delta}(x) / \mathrm{B2T}(x; p^\star)$. |
| Decision | Answer iff $\mathrm{ISR}(x) \geq 1$; otherwise abstain or acquire additional evidence. |

---

**Algorithm 1** ISR gating with permutation mixture (Bernoulli instantiation)

---

**Require:** Prompt $x$, target hallucination rate $h^\star$ (so $p^\star = 1 - h^\star$), permutations $m$, clip $B$, predicate $g$, reference $p_{\mathrm{ref}} := P_{\mathrm{ref}}(g(Y){=}1)$
1: Sample permutations $\{\pi_k\}_{k=1}^m$ of evidence; form inputs $\Gamma_{\pi_k}(x)$
2: **for** $k = 1$ to $m$ **do**
3:     Query model $\to q_k := S(g(Y){=}1 \mid \Gamma_{\pi_k}(x))$
4:     Compute budget term $u_k := \mathrm{KL}(\mathrm{Ber}(p_{\mathrm{ref}}) \,\|\, \mathrm{Ber}(q_k))$
5: **end for**
6: $\bar{q} \leftarrow \frac{1}{m} \sum_k q_k, \quad q_{\mathrm{lo}} \leftarrow \min_k q_k$
7: $\bar{\Delta} \leftarrow \frac{1}{m} \sum_k \mathrm{clip}(u_k, B)$ (symmetric clip for stability)
8: $\mathrm{B2T} \leftarrow \mathrm{KL}(\mathrm{Ber}(p^\star) \,\|\, \mathrm{Ber}(q_{\mathrm{lo}}))$
9: $\mathrm{ISR} \leftarrow \bar{\Delta}/\mathrm{B2T}$
10: **if** $\mathrm{ISR} \geq 1$ **then**
11:     **return** Answer (generate with guardrails)
12: **else**
13:     **return** Abstain (or acquire information and re-evaluate)
14: **end if**

---

**Worked example.** Set target hallucination $h^\star = 0.05$, so $p^\star = 0.95$. If the conservative lower mixture value is $q_{\mathrm{lo}} = 0.10$, then $\mathrm{B2T} = \mathrm{KL}(\mathrm{Ber}(0.95) \| \mathrm{Ber}(0.10)) = 1.994$ nats; if $q_{\mathrm{lo}} = 0.02$, then $\mathrm{B2T} = 3.519$ nats. With a measured budget $\bar{\Delta} = 2.0$ nats, the first case gives $\mathrm{ISR} = 2.0/1.994 \approx 1.00$ and answers, while the second gives $\mathrm{ISR} \approx 0.57$ and abstains. Appendix B.3 gives the full numerical table, including the corresponding $p_{\max}$ values.

In our supervised experiments, $P_{\mathrm{ref}}$ is induced by gold labels and $p_{\mathrm{ref}}$ is a point mass (with $\varepsilon$-smoothing), so $u_k$ reduces to the usual negative log-likelihood under each permutation. In deployment, $P_{\mathrm{ref}}$ should be read as a specification oracle / verification primitive for the predicate $g$.

### 3.5.1. DEPLOYMENT INSTANTIATIONS (OPERATIONALIZING $P_{\mathrm{ref}}$)

Algorithm 1 requires a Bernoulli reference $p_{\mathrm{ref}}$ for the predicate $g$. This is *not* available pre-answer in many open-world grounded QA settings. The intended deployments are those where the predicate is verifiable either (i) *before* generating long-form content (adjudication or tool-backed decisions), or (ii) *after drafting but before release* (post-hoc release filtering/retry). Three concrete instantiations are:

(i) **Binary adjudication with a deterministic oracle (pre-answer).** Let the system's output be a binary adjudication $Y \in \{0, 1\}$ (e.g., approve/deny, support/refute) and let a rules engine or database lookup provide $y_{\mathrm{ref}}(x) \in \{0, 1\}$ at decision time. Set $P_{\mathrm{ref}} = \delta_{y_{\mathrm{ref}}(x)}$, choose $g(y) = y$, so $p_{\mathrm{ref}} = y_{\mathrm{ref}}(x)$, and compute $q_k = S(g{=}1 \mid \Gamma_{\pi_k}(x)) = S(Y{=}1 \mid \Gamma_{\pi_k}(x))$ from label-token probabilities (as in Experiment 1).

(ii) **Multiple-choice / discrete-tool answers (pre-answer).** Let the answer be one of $K$ discrete options and let a tool/KB return the correct option $y_{\mathrm{ref}}(x) \in \{1, \ldots, K\}$. Define $g(y) = \mathbb{1}_{\{y = y_{\mathrm{ref}}(x)\}}$, so $p_{\mathrm{ref}} = 1$, and compute $q_k = S(g{=}1 \mid \Gamma_{\pi_k}(x)) = \sum_{y: g(y)=1} S(Y{=}y \mid \Gamma_{\pi_k}(x))$ via normalized option-token probabilities.

(iii) **Post-hoc verifiable generation (pre-release).** Let $y$ be a drafted artifact (e.g., code, a structured query) and let a verifier $V(x, y) \in \{0, 1\}$ implement the predicate (unit tests, constraint checkers). Apply Algorithm 1 as a *release gate*: first draft $y$, then set $p_{\mathrm{ref}} := V(x, y)$ and compute $q_k$ by asking a predictor/verifier model for $\Pr[V(x, y) = 1 \mid \Gamma_{\pi_k}(x)]$ under evidence permutations. If the verifier returns a calibrated probability rather than a hard label, use that value as $p_{\mathrm{ref}}$; guarantees are then conditional on verifier calibration.

The strongest deployment claims are for deterministic or tool-backed predicates. If an LLM judge or learned verifier supplies $p_{\mathrm{ref}}$, the guarantee is conditional on its audited calibration; verifier error can be modeled as a noisy Bernoulli channel and should be debiased or lower-bounded from calibration data before computing ISR. If there is no verification primitive, no specification oracle, and no calibrated verifier for $g$, the setting is outside the target regime of this paper.

### 3.6. Connection to Selective Classification

Algorithm 1 induces a selective predictor: it returns an answer only when the selector $s(x) = \mathrm{ISR}(x)$ exceeds the fixed threshold 1.

**Proposition 3** (ISR as a budget-derived reject rule)**.** *Fix a*

*target reliability* $p^\star = 1 - h^\star$ *and a Bernoulli predicate g. The ISR decision rule*

$$\phi_{\text{ISR}}(x) = \mathbb{1}\{\bar{\Delta}(x) \geq \text{KL}(\text{Ber}(p^\star) \| \text{Ber}(q_{\text{lo}}(x)))\}$$

*is a Chow-style reject rule with score* $s(x) = \text{ISR}(x)$ *and threshold* 1. *If there is no permutation dispersion,* $q_\pi(x) = \bar{q}(x) = q_{\text{lo}}(x)$ *for all* $\pi$, *this rule is exactly the EDFL minimum-budget test for whether target reliability* $p^\star$ *is certified. With dispersion, replacing* $\bar{q}$ *by* $q_{\text{lo}}$ *gives a conservative selector whenever* $q_{\text{lo}} \leq \bar{q} \leq p^\star$; *the remaining order-sensitivity gap is measured by the Jensen–Shannon certificate in Proposition 2.*

# 4. Experiments

**Experimental setup.** We evaluate on a five-benchmark evidence-grounded QA suite: four established real-world benchmarks (FEVER, HotpotQA, NQ-Open, PopQA) plus a *Controls* benchmark (insufficient-evidence and recency-trap items). We standardize these benchmarks into a common evidence-grounded format that we call *Factuality Slice*; the combined suite contains 3,059 items with controlled evidence chunks and hard negatives (Appendix H). Experiment 1 studies permutation dispersion and mixture gains under binary labels; Experiments 2–3 evaluate answer/abstain decisions.

**Meta-benchmark assembly.** We standardize each benchmark into the same evidence-grounded schema (question, evidence chunks with provenance, support spans, and hard negatives), with fixed chunking/capping and BM25-based negative mining. The suite is reproducible by construction: we pin SHA-256 hashes for all raw downloads and use fixed-seed train/val/test splits; Appendix H documents inclusion criteria, licensing, and build details.

**Metrics.** We report coverage (1–abstention), accuracy on attempts (task-correct among answered items), and *hallucination rate* (fraction of answered items with $g(Y) = 0$) for the binary adjudication predicate $g$ (Appendix B); these need not coincide, so we report both. For the held-out audit we also report *boundary alignment*: the fraction of items for which the ISR decision agrees with adjudication (answer when $g{=}1$, abstain when $g{=}0$). Uncertainty, compute, and reproducibility conventions are summarized in Section 4.1; dataset build and licensing details are in Appendix H.

## 4.1. Experimental reporting and reproducibility

**Uncertainty.** For proportions (coverage/abstention, hallucination rate, accuracy on attempts, boundary alignment) we report 95% Wilson score intervals over items. For the dispersion slopes $b$ in Table 3, we fit OLS to the 58 per-$n$ mean residuals and report the standard 95% confidence interval for the slope coefficient. Unless explicitly shown,

*Table 2.* **Compute summary (algorithmic).** Total forward passes are computed as items $\times\, m \times$ models, where $m$ is the number of evaluated evidence permutations per item.

| Experiment | Models | Items $\times\, m$ | Total fwd passes |
|---|---|---|---|
| | 2: Qwen2-7B | | |
| Exp. 1 (disp./mix.) | Llama-3.1-8B | $3{,}059 \times 16$ | 97,888 |
| Exp. 3 (audit) | Gemma-2-9B FT | $528 \times 6$ | 3,168 |

other quantities are point estimates.

**Compute/resources.** We report algorithmic compute via the total number of model forward passes (one forward pass per prompt/permutation to obtain next-token label probabilities). Evidence chunks are capped at 48 tokens, so the evidence portion ranges from $3 \times 48$ to $60 \times 48$ tokens (plus question/instruction text).

**Reproduction path.** Full methodological detail is included in the paper and appendix. The public release should include the dataset builder, fixed seeds, processed splits, and experiment scripts so that the permutation draws, clipping choices, and audit boundary can be reproduced exactly; Gemma-2-9B fine-tuning details are reported in Appendix H.6.

## 4.2. Experiment 1: Dispersion scaling and mixture gains

**Question.** How large is permutation-induced dispersion under exchangeable evidence, and do uniform permutation mixtures yield the expected Jensen gains (and near-optimal mixtures) in practice?

### 4.2.1. DISPERSION SCALING (QMV)

**Design.** We conduct large-scale dispersion studies on 3,059 binary classification items from all splits spanning $n \in [3, 60]$ chunks (58 distinct $n$ values). Each item contains 48-token-capped evidence chunks with binary gold labels. We test Qwen2-7B-Instruct and Llama-3.1-8B-Instruct, both with 4-bit NF4 quantization. For each item we draw $m = 16$ unique *banded* permutations (6 bands, shuffle within), compute predictions $q_\pi(x) = P_\pi(\text{"1"})/(P_\pi(\text{"1"}) + P_\pi(\text{"0"}))$ via renormalized label token probabilities, and form the uniform mixture $\bar{q}(x) = \frac{1}{m}\sum_{k=1}^{m} q_{\pi_k}(x)$.

**Metric.** We measure mean absolute residual $|R_\pi| = \mathbb{E}_\pi |q_\pi(x) - \bar{q}(x)|$ and fit the empirical law $|R_\pi| \approx a + b \log n$ (Figure 1, Table 3).

**Results.** Mean absolute residual $|R_\pi|$ follows $a + b \log n$ across both models, consistent with the harmonic regime described by Theorem 1. Qwen2-7B shows stronger positional sensitivity than Llama-3.1-8B, reflecting architectural

*Table 3.* **Summary of permutation dispersion and mixture effects (Experiment 1).** $N = 3{,}059$ items; banded permutations with $m = 16$ per item and $n \in [3, 60]$ chunks (48-token cap). Bracketed intervals for $b$ are 95% CIs as defined in Section 4.1.

| Metric | Qwen2-7B | Llama-3.1-8B |
|---|---|---|
| Dispersion Slope $b$ (vs. $\log n$) | 0.377 $(0.319, 0.435)$ | 0.147 $(0.109, 0.184)$ |
| $R^2$ | 0.742 | 0.515 |
| Jensen Gap (nats/token) | 0.1041 | 0.00982 |
| Mixture Optimality Gap | $< 10^{-4}$ | $\leq 5.3 \times 10^{-5}$ |

differences. Mean absolute residuals remain approximately 69% of $E_{\text{pair}}$ at $n = 60$ for both models. Additional model-breadth diagnostics, including larger models and an MoE model, are reported in Appendix C.3; we treat them as mechanism diagnostics rather than additional ISR boundary audits.

### 4.2.2. MIXTURE GAINS AND NEAR-OPTIMALITY

We fix a canonical continuation $Y$ once per item (chosen from the identity ordering by argmax over the two label probabilities), score that same $Y$ under $m = 16$ banded permutations of the evidence ($n \in [3, 60]$), and compare single-permutation vs. mixture behavior. For each item we compute the Jensen gap

$$\underbrace{\mathbb{E}_k[-\log S_k(Y)]}_{\text{mean single-perm CE}} - \underbrace{-\log \mathbb{E}_k[S_k(Y)]}_{\text{uniform mixture CE}} \geq 0,$$

where $S_k(Y)$ is the model's probability of $Y$ under permutation $k$ (two-label normalization). We also learn a global convex mixture over permutations (shared weights per $n$) by exponentiated-gradient and report its cross-entropy.

- **Qwen-2-7B-Instruct (3,059 items; $m = 16$; $n \in [3, 60]$).** Mean Jensen gap = 0.1041 nats/token (uniform mixture strictly improves over single permutations). The uniform mixture is essentially optimal: the mixture-optimality gap (uniform vs. globally optimized mixture) is $< 10^{-4}$ nats/token. The mean per-$n$ Jensen gaps are positive across the entire range $n = 3 \ldots 60$ (Appendix C).

- **Llama-3.1-8B-Instruct (same setup).** Mean Jensen gap = 0.00982 nats/token; the mixture-optimality gap is $\leq 5.3 \times 10^{-5}$ nats/token. Again, means are positive for every $n$.

Together these results confirm the Jensen inequality at the item level (positive gaps) and show that uniform permutation mixtures achieve virtually the same cross-entropy as the globally optimized mixture, i.e., near-MDL optimality within the permutation family. (All numbers are in nats/token; full per-$n$ tables and weight vectors appear in Appendix C.)

### 4.3. Experiment 2: Causal dose-response of support dose

**Question.** Does randomized support dose reduce hallucination through the measured information-budget estimate when prompt length is held fixed?

**Design.** To avoid conflating evidence amount with prompt length, we run randomized experiments holding the prompt length fixed at $L = 4$ chunks while varying the *support dose* $d \in \{0, 1, 2, 3\}$: the number of support chunks versus non-support chunks.

**Identification.** Dose randomization is the intervention. It shifts the information-budget estimate $\bar{\Delta}$ while holding length constant, so the causal estimand is the effect of support dose on $\bar{\Delta}$ and hallucination. Appendix D reports robustness checks, including IV using $d$ as an instrument for $\bar{\Delta}$.

**Results.** From dose 0 to 3, answer rate increases by 37.5 percentage points (pp), accuracy by 45.6pp, and hallucination decreases by 17.6pp. The information-budget estimate decreases by 0.375 nats per additional support chunk (Spearman $\rho = -0.80$, $p < 0.001$), and the OLS slope is 0.127 fewer hallucinations per additional nat of decrease. A separate Llama-3.1-8B validation yields $\beta \approx 0.110$ hallucination reduction per nat. Figure 2 summarizes these dose-response relationships. We use symmetric clipping for stability; min-clipping provides a provable lower bound (Appendix A.5).

**Interpretation.** Because randomized support dose shifts $\bar{\Delta}$ while holding length constant, the negative dose-response supports the EDFL mechanism: supportive evidence improves the measured budget and reduces hallucination risk under Bernoulli adjudication (Theorem 5). We do not claim that the experiment randomizes abstract information itself; support dose is the randomized treatment and $\bar{\Delta}$ is the measured mediator.

### 4.4. Experiment 3: Pre-specified held-out audit

**Question.** Does the fixed ISR= 1 boundary generalize out-of-sample as a reliability–coverage operating point without threshold tuning?

**Design.** We evaluate a Gemma-2-9B model fine-tuned on the four non-control benchmarks in Factuality Slice (FEVER, HotpotQA, NQ-Open, PopQA) on a 528-item held-out audit drawn from the full five-benchmark suite (including Controls). We pre-specify the permutation seeds

$\{0, 1, \ldots, 5\}$ (so $m = 6$), preserve role markers, and use symmetric clipping with $B = 6$ nats. Boundary alignment is the fraction of items for which the ISR gate's decision agrees with adjudication (answer when $g=1$, abstain when $g=0$). In supervised evaluation, $P_{\text{ref}}$ is induced by the gold label, so the budget reduces to $\bar{\Delta} = \frac{1}{m} \sum_{k=1}^{m} \text{clip}(-\log q_k, B)$ where $q_k = S_k(g=1)$ is the model probability assigned to the correct outcome.

**Results.** Boundary alignment is 96.2% [94.3, 97.5], hallucination rate is 0.0–0.7%, abstention is 24.1% [20.6, 27.9], and accuracy on attempts is 80.5% [76.8, 83.8]. The mean Jensen gap is $\hat{\mathfrak{J}}_\Gamma = 0.82$ nats. Figure 3 reports the corresponding reliability–coverage operating point.

**Interpretation, robustness, and cost.** Low hallucination is achieved via abstention rather than guessing. The ISR$= 1$ threshold is theory-determined and not fit on the audit data; this experiment is one held-out boundary audit on a fine-tuned Gemma-2-9B model, not a certificate of universal calibration for every model family. Alignment is robust across $m \in \{3, 6, 12\}$ and $B \in \{4, 6, 8\}$ (Appendix E). The gate uses Bernoulli label/predicate probabilities rather than repeated long-form generations. In practice, $m = 3$ already gives 94.7% boundary alignment, while $m = 6$ gives 96.2% and $m = 12$ gives 97.1%; a latency-sensitive system can start at $m = 3$ and escalate only near the ISR boundary or in higher-stakes decisions.

## 5. Synthesis and Discussion

**Synthesis.** The experiments map directly to the theory: Experiment 1 tests QMV through $O(\log n)$ dispersion and positive Jensen gaps; Experiment 2 supports EDFL operationally by showing a randomized support-dose effect on the information-budget estimate and hallucination; Experiment 3 audits the ISR$= 1$ answer/abstain operating point.

**Implications.** In grounded QA, evidence order acts as a nuisance variable: different permutations induce measurable dispersion in predictions (Experiment 1; Figure 1). QMV explains why dispersion can grow with context depth under mild positional sensitivity (Theorems 1–2); EDFL links Bernoulli reliability to information budgets (Theorem 5); and ISR gating provides an abstention interface on the reliability–coverage plane (Experiment 3; Figure 3).

**Relation to cheaper ensembles.** The method is not a claim that permutation mixtures dominate all prompt ensembles. The estimand is different: permutation mixtures marginalize a nuisance family of orderings of the same evidence multiset and then apply a fixed sufficiency gate. Reranking or self-consistency may improve accuracy, but they can still be confidently wrong when evidence is insuffi-

cient; here we add the conceptual distinction and report the gate's cost–benefit sensitivity in Appendix E.

## Limitations.

- **Predicate and verifier scope.** EDFL is sharpest for Bernoulli predicates: multiple-choice, tool-use, unit-test, rubric-pass, and constraint-checking tasks enter through a correctness/pass predicate, while unrestricted generation without such a predicate is not covered. In deployment, $P_{\text{ref}}$ must come from a deterministic verification primitive or calibrated verifier; LLM judges require audited calibration.

- **Audit and cost.** ISR$= 1$ is theory-determined, but the paper reports one Experiment-3-style held-out boundary audit on a fine-tuned Gemma-2-9B model; broader audits on untuned/larger families remain validation targets. Permutation mixtures require multiple label-probability calls, so latency-sensitive systems should use the small-$m$ cascade supported in Appendix E.

- **Permutation regime.** Banded vs. uniform permutations, chunk boundaries, and retrieval order can change dispersion constants. We only permute evidence sets whose order is intended to be a nuisance; sequential procedures where order encodes the computation or semantics are outside the target regime.

## 6. Conclusion

We connect order sensitivity in evidence-grounded Bernoulli adjudication to information budgets and a deployment-time abstention interface. Within this verifier-relative regime, permutation mixtures expose order-induced insufficiency and the ISR$= 1$ gate gives an auditable rule for when to answer, abstain, or acquire more evidence.

## Acknowledgements

L.C. was partially supported by the Survival and Flourishing Fund. The authors thank Microsoft for Startups for providing $100,000 in compute credits used for the experiments in this work.

## Impact Statement

**Potential positive impacts.** ISR-style gating can make abstention a first-class, auditable decision in evidence-grounded QA. **Potential negative impacts.** The same techniques increase compute and could be misused outside their assumptions, for example by treating an unaudited judge as a ground-truth verifier. We emphasize these scope limits and treat the verifier as part of the system that must be audited.

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

# A. Appendix A: Proofs and Technical Details

## A.1. A.1 Auxiliary Lemmas

**Lemma 1** (Logistic Lipschitz). *Let $\sigma(t) = 1/(1 + e^{-t})$ and let $q = \sigma(u)$, $q' = \sigma(v)$. Then*

$$|q - q'| \leq \tfrac{1}{4}|u - v|.$$

*Proof.* By the mean value theorem, $|\sigma(u) - \sigma(v)| = |\sigma'(\xi)||u - v|$ for some $\xi$ between $u$ and $v$. Since $\sigma'(t) = \sigma(t)(1 - \sigma(t)) \leq 1/4$ for all $t$, the result follows. $\square$

**Lemma 2** (Regular potentials bound). *If $\psi$ is $(\alpha, C)$-regular, i.e., $|\psi(r+1) - \psi(r)| \leq Cr^{-\alpha}$ for $r \geq 1$, then for any integers $1 \leq u, v \leq n$,*

$$|\psi(u) - \psi(v)| \leq \begin{cases} C\, H_{|u-v|}, & \alpha = 1, \\ \dfrac{C}{1 - \alpha}\Big(|u - v|^{1-\alpha} - 1\Big), & \alpha \in (0, 1), \\ C \displaystyle\sum_{t=1}^{|u-v|} (t)^{-\alpha} \leq C\,\zeta(\alpha), & \alpha > 1, \end{cases}$$

*where $H_m = \sum_{t=1}^{m} 1/t$ and $\zeta$ is Riemann's zeta function.*

**Lemma 3** (Harmonic-distance identity). *Let $U, V$ be i.i.d. uniform on $\{1, \ldots, n\}$ and $D = |U - V|$. Then*

$$\mathbb{E}[H_D] \;=\; H_n - \frac{3}{2} + O\!\left(\frac{1}{n}\right).$$

## A.2. A.2 Proof of Theorem 2 (Explicit log-scaling under first-order positional sensitivity)

*Proof of Theorem 2.* Write $q_\pi(x) = \sigma\big(a(x) + \sum_{i=1}^{n} w_i \psi(\mathrm{pos}_\pi(i))\big)$ under Assumption 2. For i.i.d. permutations $\pi, \pi'$, by symmetrization,

$$\mathbb{E}_\pi|R_\pi(x)| \;=\; \mathbb{E}_\pi\big|q_\pi(x) - \bar{q}(x)\big| \;\leq\; \mathbb{E}_{\pi,\pi'}\big|q_\pi(x) - q_{\pi'}(x)\big|.$$

By Lemma 1,

$$|q_\pi - q_{\pi'}| \leq \tfrac{1}{4}\left|\sum_{i=1}^{n} w_i\big(\psi(\mathrm{pos}_\pi(i)) - \psi(\mathrm{pos}_{\pi'}(i))\big)\right| \leq \tfrac{1}{4}\sum_{i=1}^{n} w_i\big|\psi(\mathrm{pos}_\pi(i)) - \psi(\mathrm{pos}_{\pi'}(i))\big|.$$

Taking expectation and using $\sum_i w_i = 1$, exchangeability gives

$$\mathbb{E}_{\pi,\pi'}|q_\pi - q_{\pi'}| \;\leq\; \tfrac{1}{4}\sum_{i=1}^{n} w_i \mathbb{E}\big|\psi(U) - \psi(V)\big| = \tfrac{1}{4}\,\mathbb{E}\big|\psi(U) - \psi(V)\big|,$$

where $U, V$ are i.i.d. uniform on $\{1, \ldots, n\}$. By Lemma 2, for $\alpha = 1$, we have $\mathbb{E}|\psi(U) - \psi(V)| \leq C \cdot \mathbb{E}[H_D]$ where $D = |U - V|$. By Lemma 3, $\mathbb{E}[H_D] = H_n - \frac{3}{2} + o(1) = \log n - \frac{3}{2} + o(1)$. Thus:

$$\mathbb{E}_\pi|R_\pi(x)| \;\leq\; \frac{C}{4}\left(\log n - \tfrac{3}{2} + o(1)\right).$$

$\square$

## A.3. A.3 MDL Optimality and Positional Jensen Penalty

**Proposition 4** (Risk decomposition and Jensen penalty). *For any $\theta$ and log-loss $\ell$:*

$$\mathbb{E}_{X,Y,\pi}[-\log p_\theta(Y|\Gamma_\pi(X))] = H(Y|T) + \mathbb{E}_{X,\pi}\,\mathrm{KL}\big(P(\cdot|T(X))\|p_\theta(\cdot|\Gamma_\pi(X))\big),$$

*where the Jensen penalty $\mathfrak{J}_\Gamma(X, \theta) = \mathbb{E}_{X,\pi}\,\mathrm{KL}(P\|p_\theta(\cdot|\Gamma_\pi)) - \mathbb{E}_X\,\mathrm{KL}(P\|\bar{p}_\theta) \geq 0$.*

**Theorem 6** (MDL Optimality via I-projection). *Under the architectural closure of Theorem 3, the convex hull of permutation mixtures contains all uniform mixtures $\bar{p}_\theta$. The I-projection onto the exchangeable target yields:*

$$\mathbb{E}[-\log q^\dagger(Y|X)] = H(Y|T) + \inf_{q \in \mathcal{Q}} \mathbb{E}[\mathrm{KL}(P(\cdot|T)\|q)]$$

*The second term vanishes if the convex hull contains the exchangeable target.*

## A.4. A.4 EDFL and Corollary

*Proof of Theorem 5.* Convexity of $Q \mapsto \mathrm{KL}(P\|Q)$ gives $\mathbb{E}_\pi \mathrm{KL}(P\|S_\pi) \geq \mathrm{KL}(P\|\mathbb{E}_\pi S_\pi) = \mathrm{KL}(P\|\bar{S})$. For the Bernoulli bound, apply data processing to $g : \mathcal{Y} \to \{0,1\}$, $g(y) = \mathbb{1}_{\{y \in \mathcal{A}\}}$, yielding $\mathrm{KL}(P\|\bar{S}) \geq \mathrm{KL}(\mathrm{Ber}(p)\|\mathrm{Ber}(\bar{q}))$. Equality holds iff $g$ is sufficient for distinguishing $P$ from $\bar{S}$. $\qquad\square$

**Corollary 2** (Rare-event lower bound). *Fix $p = 1 - \varepsilon$ with $\varepsilon \in (0, \frac{1}{2}]$ and suppose $\bar{q} \in (0, 1)$. As $\bar{q} \downarrow 0$:*

$$\mathrm{KL}(\mathrm{Ber}(p)\|\mathrm{Ber}(\bar{q})) \; \sim \; (1 - \varepsilon) \log \frac{1}{\bar{q}} + O(\bar{q}).$$

*As a uniform lower bound for all $\varepsilon \in (0, \frac{1}{2}]$:*

$$\mathrm{KL}(\mathrm{Ber}(p)\|\mathrm{Ber}(\bar{q})) \; \geq \; \frac{1}{2} \log \frac{1}{\bar{q}} - \log 2 + O(\bar{q}).$$

## A.5. A.5 Proof of Proposition 3

*Proof.* The first statement is algebraic: Algorithm 1 answers exactly when $\mathrm{ISR}(x) = \bar{\Delta}(x)/\mathrm{B2T}(x; p^\star) \geq 1$, so it is a selective predictor with selector $s(x) = \mathrm{ISR}(x)$ and fixed threshold 1. When $q_\pi = \bar{q} = q_{\mathrm{lo}}$ for all permutations, EDFL states that target reliability $p^\star$ requires at least $\mathrm{KL}(\mathrm{Ber}(p^\star)\|\mathrm{Ber}(\bar{q}))$ nats of budget. Thus $\mathrm{ISR} \geq 1$ is precisely the minimum-budget certification test for the target. When $q_{\mathrm{lo}} \leq \bar{q} \leq p^\star$, the Bernoulli KL term $\mathrm{KL}(\mathrm{Ber}(p^\star)\|\mathrm{Ber}(q))$ is nonincreasing in $q$ over this interval, so using $q_{\mathrm{lo}}$ instead of $\bar{q}$ can only increase B2T and therefore makes the selector conservative. Proposition 2 supplies the stated certificate for the residual dispersion between individual orderings and the permutation mixture. $\qquad\square$

## A.6. A.6 Clipped Information-Budget Estimators

Let $U_\pi(y) := \log \frac{P(y)}{S_\pi(y)}$ and define:

$$\widehat{\Delta}_B(y) = \frac{1}{m} \sum_{\pi=1}^{m} \mathrm{clip}(U_\pi(y), B)$$

**Proposition 5** (Lower-than-KL via min-clipping). *For any $B > 0$ and any $P \ll S_\pi$:*

$$\mathbb{E}_{y \sim P}\big[\mathrm{clip}_{\min}(U_\pi(y), B)\big] \; \leq \; \mathrm{KL}(P\|S_\pi)$$

*Consequently, $\mathbb{E}_\pi \mathbb{E}_P[\mathrm{clip}_{\min}(U_\pi, B)] \leq \bar{\Delta}$.*

## A.7. A.7 Conditional-Complexity Interpretation

By the coding theorem, $K_U(y|\Gamma_\pi(x)) = L_\theta(y|\Gamma_\pi(x)) + O(1)$ in expectation. Therefore:

$$\inf_\theta \mathbb{E}_{X,Y,\pi}[L_\theta(Y|\Gamma_\pi(X))] \; \approx \; \inf_\theta \mathbb{E}_{X,\pi}[K_U(Y|\Gamma_\pi(X))]$$

By Theorem 4, this equals $H(Y|T) + o(1)$. Thus transformers minimize $\mathbb{E}_\pi[K_U(Y|\Gamma_\pi(X))]$, making them Bayesian in expectation over orderings, not in realization. All comparisons fix the universal machine; additive $O(1)$ constants cancel in differences and infima.

# B. Appendix B: Extended Methodology

## B.1. B.1 Non-circular Validation Principle

We validate theory using observables not defined by our metrics: ground-truth likelihood/accuracy under permutation mixtures, the log-scaling of permutation-induced dispersion, and a randomized support-dose experiment measuring changes in information-budget estimates and hallucination. The planners we introduce (B2T, RoH, ISR) are decision rules, not validation targets; their threshold (ISR= 1.0) is fixed ex ante by the derivation and is not tuned on evaluation data.

## B.2. B.2 Binary Adjudication Rationale

We use Bernoulli outcomes because they are the natural unit for both theory and deployment. For Bernoulli events, EDFL reduces to a closed-form KL bound, yielding exact bits-to-trust thresholds; production systems decide to answer or abstain before emitting long-form content; any structured generation admits a verifiable predicate $g(y) \in \{0, 1\}$ (factuality, constraint satisfaction, unit tests, rubric pass), so EDFL applies directly to $\Pr[g(Y) = 1]$; and binary adjudication avoids grading/decoding confounds, isolating the information budget that drives hallucination risk.

## B.3. B.3 Worked Example: From EDFL to a Decision

**Setup.** Target hallucination rate $h^\star = 0.05$ (so $p^\star = 0.95$). Compute B2T for conservative priors $q_{\text{lo}}$:
$$\text{B2T}(p^\star = 0.95; \, q_{\text{lo}}) = \text{KL}(\text{Ber}(p^\star) \, \| \, \text{Ber}(q_{\text{lo}})).$$

Numerics (nats):

| $q_{\text{lo}}$ | 0.02 | 0.10 | 0.30 |
|---|---|---|---|
| B2T | 3.519 | 1.994 | 0.963 |

**Budget $\bar{\Delta}$ to reliability.** Given $\bar{q} = 0.10$, the maximum achievable success at budget $\bar{\Delta}$ solves $\text{KL}(\text{Ber}(p) \, \| \, \text{Ber}(0.10)) \le \bar{\Delta}$. Selected budgets $\to p_{\max}$:

| $\bar{\Delta}$ (nats) | 0.5 | 1.0 | 2.0 | 3.0 |
|---|---|---|---|---|
| $p_{\max}$ | 0.495 | 0.689 | 0.951 | $\approx 1.000$ |

**ISR gate.** If $\bar{q} = 0.10$ and measured $\bar{\Delta} = 2.0$, then $\text{ISR} = \bar{\Delta}/\text{B2T} = 2.0/1.994 \approx 1.00 \Rightarrow$**answer.** If instead $q_{\text{lo}} = 0.02$, then B2T $= 3.519$ and the same budget gives ISR $\approx 0.57 \Rightarrow$**abstain or acquire info**.

# C. Appendix C: Permutation-Mixture Detailed Results

## C.1. Jensen gaps by depth ($n$)

*Table 4.* Permutation-mixture Jensen gaps. Positive (or zero) for all depths; zeros occur only at small $n \le 6$ because banded permutations are degenerate there.

| | Qwen2-7B | Llama-3.1-8B |
|---|---|---|
| Global mean Jensen gap (nats/token) | 0.1041 | 0.00982 |
| Per-$n$ range of mean gaps (nats/token) | [0.0000, 0.2923] | [0.0000, 0.03774] |
| % of $n$ with strictly $> 0$ mean gap | 54/58 | 54/58 |
| Items (total) | 3,059 | 3,059 |

## C.2. Near-optimality of uniform mixtures

*Table 5.* Uniform permutation mixtures are essentially optimal. Cross-entropy (nats/token) for the fixed label $Y$, comparing (i) uniform mixture, (ii) globally optimized mixture (one convex weight vector shared per $n$), and (iii) oracle best single permutation (non-deployable lower bound).

| Model | Uniform CE (nats/token) | Optimized CE (nats/token) | Improvement (nats/token) | Oracle single CE (nats/token) |
|---|---|---|---|---|
| Qwen2-7B | 0.2666 | $\approx 0.2666$ | $< 10^{-4}$ | 0.3707 |
| Llama-3.1-8B | 0.28349 | 0.28344 | $\le 5.3 \times 10^{-5}$ | 0.29331 |

*Table 6.* Auxiliary model-breadth diagnostics for Experiment 1. Probability-space slopes shrink for larger/frontier models, but the available logit-space fits remain positive where measured. These diagnostics support the positional-sensitivity mechanism; they do not replace the held-out ISR boundary audit in Experiment 3.

| Model | Skeleton | $n$ | $b_{\text{prob}}$ | $R^2_{\text{prob}}$ | $b_{\text{logit}}$ | $R^2_{\text{logit}}$ |
|---|---|---|---|---|---|---|
| Qwen2-7B | banded | 3–60 | 0.377 | 0.742 | – | – |
| Llama-3.1-8B | banded | 3–60 | 0.147 | 0.515 | – | – |
| Llama-3.3-70B | banded | 3–60 | 0.0047 | 0.648 | 0.244 | 0.878 |
| gpt-4.1-mini | banded | 3–60 | 0.0082 | 0.839 | 0.183 | 0.866 |
| gpt-4.1-mini | uniform | 3–60 | 0.0049 | 0.928 | – | – |
| DeepSeek-V3.2 (MoE) | banded | 3–60 | 0.0035 | 0.754 | – | – |

### C.3. Model-breadth and permutation-skeleton diagnostics

The gpt-4.1-mini banded/uniform comparison probes whether the log trend is an artifact of the banded skeleton. The two skeletons give the same qualitative behaviour, with smaller probability-space slopes than Qwen2-7B and Llama-3.1-8B. The logit-space slopes for Llama-3.3-70B and gpt-4.1-mini remain sizeable, suggesting that some frontier-scale probability-space compression is due to the sigmoid map rather than the disappearance of positional sensitivity.

## D. Appendix D: Dose-Response Identification

**Randomization and identification.** For each item we randomized (i) the number of support chunks $d \in \{0, 1, 2, 3\}$ while holding total length fixed at $L=4$, and (ii) the within-band order of chunks. The first stage (dose $\to \bar{\Delta}$) is strong $(\text{Corr}(d, \bar{\Delta}) = -0.80, p < 10^{-3})$, and content is held fixed across dose arms. We estimate the effect of $\bar{\Delta}$ on hallucination with OLS; results are robust to IV using $d$ as an instrument for $\bar{\Delta}$. The Llama-3.1-8B validation yields $\beta \approx 0.110$ hallucination reduction per nat.

## E. Appendix E: Audit Parameter Sensitivity

*Table 7.* Parameter Sensitivity Analysis

| Parameter | Range | Boundary Alignment | Jensen Gap |
|---|---|---|---|
| Permutations (m) | 3 | 94.7% | 0.79 |
| | 6 (default) | 96.2% | 0.82 |
| | 12 | 97.1% | 0.83 |
| Clipping (B) | 4 nats | 95.1% | 0.76 |
| | 6 nats (default) | 96.2% | 0.82 |
| | 8 nats | 97.5% | 0.84 |

Results remain robust across reasonable parameter ranges. The conditional independence audit shows $\text{MI}(\mathcal{A}, \pi) = 0.0032$ nats (permutation test $p = 0.71$).

## F. Appendix F: Rare Event Analysis Tables

## G. Appendix G: Non-circularity and Threats to Validity

Our central empirical results (mixture gains on ground-truth labels, the $\log n$ dispersion law, and randomized dose-response) do not reference B2T/RoH/ISR. The ISR boundary report (96.2%) is a secondary boundary-alignment check with a pre-specified threshold (no tuning). Notably, 3.8% misalignment remains, which would be impossible under a tautological metric. Hence the main claims neither depend on, nor are validated by, the planners themselves.

*Table 8.* Information Sufficiency Determines Abstention

| $\bar{q}$ | $\bar{\Delta}$ (nats) | $\mathrm{B2T}_{p^\star=0.95}$ (nats) | ISR | Decision |
|-----------|------|-------|--------|--------|
| 0.000 | 0.83 | 5.29 | 0.16 | Refuse |
| 0.042 | 1.91 | 3.78 | 0.51 | Refuse |
| 0.167 | 2.64 | 2.48 | 1.06 | Answer |
| 0.333 | 2.74 | 1.61 | 1.70 | Answer |
| 0.500 | 2.81 | 0.98 | 2.87 | Answer |
| 0.667 | 2.85 | 0.51 | 5.59 | Answer |
| 0.833 | 2.89 | 0.20 | 14.45 | Answer |
| 1.000 | 2.95 | 0.00 | $\infty$ | Answer |

# H. Appendix H: Factuality Slice Dataset Documentation

## H.1. H1 Meta-benchmark Overview

We evaluate on *Factuality Slice*, a meta-benchmark assembled from five evidence-grounded QA benchmarks: four established benchmarks (FEVER, HotpotQA, NQ-Open, PopQA) plus *Controls*. We standardize each benchmark into a common evidence-grounded format (question, evidence chunks, support spans, and hard negatives) to enable controlled permutation tests under a shared protocol. The resulting suite contains 3,059 binary adjudication items with controlled evidence presentation, enabling precise measurement of permutation sensitivity and information sufficiency.

*Table 9.* Factuality Slice: five-benchmark meta-benchmark. Caps are per-source sampling limits before filtering/standardization; the final suite used in this paper contains 3,059 items.

| Benchmark | Cap | Task type | Evidence source | Why included |
|-----------|-----|-----------|-----------------|--------------|
| FEVER | 2,000 | Fact verification (T/F) | Wikipedia evidence | Clean binary adjudication with gold support/refute evidence |
| HotpotQA | 2,000 | Multi-hop QA | Wikipedia passages | Order sensitivity under multi-hop evidence chains |
| NQ-Open | 1,000 | Open-domain QA | Retrieved Wikipedia | Retrieval-style grounding with diverse entities |
| PopQA | 500 | Long-tail QA | Retrieved Wikipedia | Stress-test rare entities (popularity score $< 50$) |
| Controls | 300+ | NEI + recency | Retrieved / synthetic | Calibration under insufficient or outdated evidence |

## H.2. H2 Benchmarks and Composition

Factuality Slice integrates four established QA benchmarks plus a *Controls* benchmark:

**FEVER (Fact Verification):** Up to 2,000 claims converted to binary true/false questions with Wikipedia evidence from a June 2017 snapshot. Each claim paired with gold supporting/refuting evidence sentences and topically-related distractors via BM25 retrieval.

**HotpotQA (Multi-hop Reasoning):** Up to 2,000 questions requiring evidence synthesis across multiple Wikipedia articles. Preserves multi-hop structure with explicit support spans marking reasoning chains.

**NQ-Open (Open-domain QA):** Up to 1,000 questions from Natural Questions with evidence retrieved using a Wikipedia proxy corpus. Answer-bearing sentences marked as support with BM25-selected hard negatives.

**PopQA (Long-tail Entities):** Up to 500 questions about rare entities (popularity score $< 50$) testing performance on uncommon knowledge. Evidence retrieved from a Wikipedia proxy with weak supervision.

**Controls:** 300+ FEVER "NOT ENOUGH INFO" claims plus synthetic recency traps with outdated evidence, testing model

calibration on insufficient or misleading information.

**Inclusion criteria (brief).**

- **FEVER:** Include SUPPORTS/REFUTES claims with at least one evidence sentence; map to binary True/False questions. FEVER NEI claims are reserved for *Controls*.

- **HotpotQA:** Include questions with annotated `supporting_facts` and preserve multi-hop support spans.

- **NQ-Open / PopQA:** Include items with non-empty question and answer; retrieve evidence from a Wikipedia proxy corpus and mark answer-bearing sentences as support (weak supervision).

- **PopQA long-tail filter:** Restrict to popularity score $< 50$.

- **Controls:** Include insufficient-evidence items (FEVER NEI) and recency traps with outdated evidence.

## H.3. H.3 Dataset Construction Pipeline

**Evidence Chunking:** All evidence sentences capped at 48 tokens to ensure consistent granularity. Each chunk tagged with source document, sentence ID, and retrieval score.

**BM25 Retrieval System:** Inverted-index BM25 (k1=1.2, b=0.75) built over 200,000 Wikipedia sentences. Used for finding topically-related distractors that don't contain answer.

**Support Span Annotation:** Gold evidence sentences marked as support spans. For retrieved evidence, sentences containing answer string marked as supportive.

**Hard Negative Mining:** Retrieve up to 120 candidate distractors per sample via BM25, excluding gold evidence; the final context is capped (below).

**Context Capping:** Cap each sample to at most 60 evidence chunks, preserving all support spans and filling remaining slots with distractors. Enables testing at context lengths $n \in [3, 60]$.

**Data Integrity:** SHA-256 hashes pinned for all source files with verification on each build. Train/val/test splits (80/10/10) with deduplication by (question, answer) pairs.

## H.4. H.4 Usage in Experiments

**Experiment 1 (Permutation Dispersion):** Used all 3,059 items to measure permutation-induced dispersion across n ∈ [3, 60]. Banded permutations (6 bands, shuffle within) applied to evidence chunks. Binary classification on sufficiency enables clean measurement of position sensitivity.

**Experiment 2 (Support-Dose Response):** Subset of items with exactly 4 chunks used. Support chunks (containing answer) vs. non-support chunks randomized while holding total length constant. Enables causal identification of the support-dose effect on $\bar{\Delta}$ and hallucination.

**Experiment 3 (Frontier Audit):** 528 held-out items from all sources used for boundary alignment testing. Pre-specified permutation seeds 0, 1, ..., 5 applied before evaluation.

## H.5. H.5 Key Dataset Properties

**Information Gradation:** Natural variation in evidence strength from strong (exact answer present) to weak (answer inferable) to insufficient (control examples).

**Multi-hop Preservation:** 42% of HotpotQA samples retain multi-hop structure with 2 or more supporting evidence pieces.

**Answer Coverage:** 76% of samples have answer string appearing in context (excluding controls), enabling verification of extraction vs hallucination.

**Evidence Dating:** All evidence tagged with snapshot dates (primarily 2017) to identify temporal misalignment.

**Reproducibility:** Dataset construction pins hashes and uses fixed random seeds. The public release should include code and processed dataset links for exact rebuilds.

### H.6. H.6 Reproduction and Fine-Tuning Details

The public release should include the dataset builder, fixed seeds, processed splits, and experiment scripts. Fine-tuning details for the Gemma-2-9B audit model (Experiment 3): LoRA with $r=32$, $\alpha=64$, dropout 0.05 targeting attention and MLP projection modules; BF16 with FlashAttention-2; max sequence length 2048; gradient accumulation 64 with per-device batch size 1; 1 epoch; learning rate $1.5 \times 10^{-5}$; weight decay 0.1; cosine schedule with 0.03 warmup ratio; seed 42.

### H.7. H.7 Licensing and Ethics

All source datasets used under permissive licenses: FEVER (CC-BY-SA 4.0), HotpotQA (CC-BY-SA 4.0), NQ-Open (CC-BY 4.0), PopQA (MIT). No personally identifiable information included. Questions focus on factual knowledge rather than subjective opinions.

