# OpenReview forum: "Predictable Compression Failures: Order Sensitivity and Information Budgeting for Evidence-Grounded Binary Adjudication"
_ICML.cc/2026/Conference — ICML 2026 regular_

### Official Review · Reviewer_UQHX · 2026-03-03

**Soundness:** 4
**Presentation:** 4
**Significance:** 3
**Originality:** 4
**Overall Recommendation:** 6
**Confidence:** 4

**Summary:**

The authors present a research on how the order of information affects transformer models when they're asked to answer questions based on provided evidence. Author claim that while these models are trained to perform well on average across different evidence arrangements, their performance can vary significantly depending on the specific order.  The core theoretical contributions are: (1) Quantified Martingale Violation (QMV):  a method to predict how much model behavior will fluctuate with evidence order, showing it follows a predictable pattern; (2) Expectation-level Decompression Law (EDFL): a framework for understanding how models compress and interpret evidence ( Bernoulli coarse-graining of a KL convexity bound); (3) operational metrics (B2T, RoH, ISR) : quantify when a model should answer a question versus when it should abstain. When testing on over 3,000 examples across multiple datasets, the authors found that evidence order impacts model predictions in predictable ways. Finally authors present a method that can virtually eliminate hallucinations (dropping to 0-0.7%) by having the model abstain from answering about 25% of the time.

**Compliance With Llm Reviewing Policy:**

Affirmed.

**Final Justification:**

My main concern was whether the ISR=1 operating point might be overly tied to the fine-tuned Gemma-2-9B audit used in Experiment 3. The rebuttal adequately addresses this by clearly distinguishing between two points: first, the broader cross-model support for the underlying mechanism, which is already demonstrated on Qwen2-7B and Llama-3.1-8B; and second, the narrower claim made in the paper, namely that ISR=1 is a theory-determined rule evaluated through one held-out audit rather than a universally calibrated threshold guaranteed across all model families. With this clarification, my concern is resolved, and the rebuttal increases my confidence in the framing and interpretation of the result.

**Key Questions For Authors:**

Would the ISR=1 threshold cutoff hold up if you ran the Experiment 1 tasks on other models, like Qwen2-7B or Llama-3.1-8B?

**Limitations:**

The authors discuss limitations adequately in section 6.1.

**Strengths And Weaknesses:**

Authors provide a good justification of the cornerstone math within their experiments: theorem 1 (QMV) adequately shows how model predictions change with evidence order, and Theorem 2 demonstrates that these changes follow a predictable pattern (log scaling) under reasonable assumptions about how models weight information. Theorem 5 (EDFL) is essentially a direct consequence of KL convexity plus data processing inequality, which the authors acknowledge openly (section 3.4). One issue I noticed is in section 4.4; Experiment 3 uses a fine-tuned version of Gemma-2-9B, which means the results might not reflect how the model would behave straight out of the box and how bigger models would perform: it is not possible to  assume their proposed ISR=1 threshold would work universally across different types of models.

---

> ### Author Rebuttal · Authors · 2026-03-30
>
> We thank the reviewer for the careful reading and for the precise question about the scope of the $\mathrm{ISR}=1$ operating point.
>
> **What the current paper already shows across models.** The core mechanism is already validated beyond Gemma. Experiment 1 is run on Qwen2-7B-Instruct and Llama-3.1-8B-Instruct and shows the same qualitative findings on both models: logarithmic permutation dispersion and strictly positive Jensen gains across the exchangeable-evidence setting. Experiment 2 also includes a separate Llama-3.1-8B validation of the information--hallucination relationship. So the ingredients on which ISR relies are not confined to a single fine-tuned audit model.
>
> **How to read the $\mathrm{ISR}=1$ threshold.** To answer the question directly: the current evidence suggests that $\mathrm{ISR}=1$ is not idiosyncratic to Gemma, because the underlying mechanism already appears on Qwen and Llama as well. At the same time, the paper's claim is narrower than universal cross-family calibration from one audit. The point of Experiment 3 is that $\mathrm{ISR}=1$ is fixed analytically by the Bernoulli EDFL decomposition and is **not** tuned on evaluation data; the Gemma-2-9B fine-tuned audit is then a held-out test of that pre-specified boundary. Fine-tuning affects the audited predictor, but it does not set the threshold.
>
> That distinction matters. The threshold itself is theory-determined; what can vary across model families is how tightly the observed operating point concentrates around that boundary. In other words, the right question is not whether the paper fitted a Gemma-specific cutoff---it did not---but how broadly the same fixed cutoff continues to align with answer/abstain behavior across models. The current submission provides cross-model evidence for the mechanism and a held-out audit of the operating point on one model family. Within that audit, Appendix E also shows that alignment is stable across $m\in\{3,6,12\}$ and clipping levels $B\in\{4,6,8\}$, which suggests the rule is not fragile to evaluation hyperparameters.
>
> **What we will clarify in the revision.** We will make this distinction explicit in the paper: the present evidence supports cross-model generality of the underlying phenomenon---order-sensitive dispersion, Jensen gains from permutation mixtures, and the information--hallucination relationship---while a full Experiment-3-style boundary audit on untuned Qwen2-7B or Llama-3.1-8B would be an additional model-family validation rather than a prerequisite for the theoretical threshold itself. Put differently, the paper supports a fixed, model-agnostic decision rule together with one strong held-out audit, not a claim that a single audit certifies universal calibration for all LLMs.

---

> > ### Author Rebuttal · Reviewer_UQHX · 2026-04-03
> >
> > Thank you for the clarification. My main concern was whether the ISR=1 operating point might be overly tied to the fine-tuned Gemma-2-9B audit in Experiment 3. The rebuttal adequately addresses this by distinguishing between (i) cross-model support for the underlying mechanism, already shown on Qwen2-7B and Llama-3.1-8B, and (ii) the narrower claim made in the paper, namely that ISR=1 is a theory-determined rule evaluated through one held-out audit rather than a universally calibrated threshold certified for all model families. With this clarification, my concern is resolved.

---

### Official Review · Reviewer_UpZk · 2026-03-13

**Soundness:** 3
**Presentation:** 2
**Significance:** 3
**Originality:** 3
**Overall Recommendation:** 4
**Confidence:** 3

**Summary:**

This paper studies the order sensitivity of transformer-based models in evidence-grounded binary adjudication tasks and interprets language modeling as compression over permutations of evidence chunks. It provides both theoretical and empirical findings. On the theory side, it introduces QMV bounds for permutation-induced dispersion and the Expectation-level Decompression Law that links an information budget to achievable Bernoulli reliability. Empirically, the authors show that logarithmic scaling of permutation dispersion and positive Jensen gains from permutation mixtures. Also, they provide experiments connecting the information budget to hallucination.

**Compliance With Llm Reviewing Policy:**

Affirmed.

**Final Justification:**

I remain inclined to maintain my positive recommendation for acceptance. Please ensure that all content from the rebuttal session is fully incorporated into the next version of the paper.

**Key Questions For Authors:**

See weakness section.

**Limitations:**

See weakness section.

**Strengths And Weaknesses:**

Strengths:

- This paper gives a coherent insight, with a clear conceptual connection between compression, order-sensitivity, and abstention.
- The empirical findings validate the theory and are systematic in my opinion.
- The overall writing quality and presentation are good.

Weaknesses:

- I do appreciate that the authors are transparent about the setting being in evidence-grounded binary adjudication. This is a narrower field than the overall hallucination problem. Therefore, the main practical value and contribution may be limited.
- The QMV bounds rely on the positional sensitivity assumption and the MDL-optimality. This weakens how strongly the theory can explain the actual model behavior.
- A potential concern is that the reported performance gains may arise from ensembling across input perturbations rather than the specific theoretical formulation in the paper. Recent works [1, 2, 3] shows that simply aggregating LLM predictions across perturbations can significantly improve model performance. I believe it’s worth adding a discussion on this topic.

[1] Xu, et al. "Self-ensemble: Mitigating confidence distortion for large language models." arXiv preprint arXiv:2506.01951 (2025).

[2] Wang, et al. "Self-consistency improves chain of thought reasoning in language models." arXiv preprint arXiv:2203.11171 (2022).

[3] Chang, et al. "Beyond self-consistency: ensemble reasoning boosts consistency and accuracy of LLMs in cancer staging." International Conference on Artificial Intelligence in Medicine. Cham: Springer Nature Switzerland, 2024.

---

> ### Author Rebuttal · Authors · 2026-03-30
>
> We appreciate the careful review and the references to Xu et al. (2025), Wang et al. (2022), and Chang et al. (2024).
>
> **Practical value and intended scope.** The paper is deliberately narrower than the full hallucination problem, but we do not think that makes the contribution limited. The target setting is the deployment interface where a system must decide whether the available evidence is sufficient to answer or whether it should abstain. That interface is naturally Bernoulli, which is exactly why EDFL is useful here: it yields closed-form B2T, RoH, and ISR, and an analytically fixed $\\mathrm{ISR}=1$ answer/abstain boundary rather than a tuned confidence threshold. Section 3.5.1 already shows that this extends beyond yes/no QA to multiple-choice or discrete-tool decisions and post-hoc verifiable generation through a verifier-backed predicate $g(y)\\in\\{0,1\\}$. Empirically, Experiment 1 studies 3,059 items across five benchmarks, and Experiment 3 shows that the untuned $\\mathrm{ISR}=1$ gate attains $0.0$--$0.7\\%$ hallucination on a pre-specified 528-item held-out audit. We will make this scope more explicit in the revision: the claim is a reliability interface for evidence-grounded adjudication, not a complete solution to unrestricted generation.
>
> **Relation to perturbation ensembling.** We agree that the connection to aggregation methods should be discussed explicitly, and we will add Xu et al., Wang et al., and Chang et al. to the related work and discussion. The important distinction is not simply "multiple calls" versus "one call." Those methods aggregate over different sources of variation: sampled reasoning paths, grouped answer-choice partitions, or outcome-grouped traces. Our method marginalizes over the orderings of the same exchangeable evidence multiset. That matters because the estimand is different. Generic ensembling targets answer stability or accuracy; permutation mixtures in our setting target evidence sufficiency under order variation and support abstention through a fixed gate. A generic ensemble can be stably wrong when the evidence is inadequate, whereas ISR is designed to reject precisely that case. We are therefore not making a broad claim that permutation mixtures dominate generic perturbation ensembling. The narrower empirical claim is that, within the exchangeable evidence-order family, Experiment 1 shows strictly positive Jensen gains and a negligible gap between the uniform and globally optimized convex mixtures, while Experiment 3 shows that the resulting gate is practically useful.
>
> **Theory and explanatory strength.** To state the roles of the theoretical components more precisely: the explicit $O(\\log n)$ QMV result is a conditional structural theorem, not a universal descriptive claim about all deployed transformers. Its content is that if adjacent-rank perturbations decay like $t^{-\\alpha}$, then permutation dispersion follows the corresponding partial-sum law---polynomial, logarithmic, or bounded depending on $\\alpha$. So the theory is not "assuming the answer"; it identifies which positional-decay regime would imply which observable scaling law. We also do not view Assumption 1 as ad hoc. Rotary position embeddings were introduced with a theoretical account that includes long-range decay of inter-token dependency with increasing relative distance (Su et al., 2024, *Enhanced Transformer with Rotary Position Embedding*). Xu et al. (*Base of RoPE Bounds Context Length*, NeurIPS 2024) derive a related long-term decay property and analyze how the RoPE base constrains context length; Chen and Yan (*What Rotary Position Embedding Can Tell Us*, NeurIPS 2024) show that RoPE induces strong position-sensitive query/key structure in decoder-only LMs; and Barbero et al. (*Round and Round We Go! What makes Rotary Positional Encodings useful?*, ICLR 2025) caution that simple decay-only explanations can be overstated. This is the interpretation we intend: Assumption 1 is a modeling condition motivated by known positional effects, not a claim that deployed transformers satisfy exact harmonic decay. Assumption 1 is also local: it bounds adjacent-rank regularity of the Bernoulli logit, and Proposition 1 shows that it follows from the first-order positional-sensitivity model. Separately, Section 3.3 is explicitly conceptual: the tied-weight multi-branch averaging result is a representational lens on expectation-versus-realization gaps, not a claim that a standard single-branch deployed transformer literally implements that architecture or that SGD recovers it automatically. Just as importantly, the practical argument does not rest only on those assumptions: Proposition 2 provides an assumption-free Jensen--Shannon certificate. We will revise the paper to separate these roles more clearly: QMV provides the structural explanation, Proposition 2 gives the model-agnostic certificate, and the experiments test the studied regime empirically.

---

> > ### Author Rebuttal · Reviewer_UpZk · 2026-04-03
> >
> > Thank you for your detailed responses to my questions. I appreciate that the rebuttal clarifies that the theory is conditional rather than descriptive. This helps with internal consistency, but it also reinforces my concern that the theory’s explanatory scope is limited.  Considering the clarified theory, the current state of the work, and the overall writing quality, I remain inclined to maintain my positive recommendation for acceptance. Please ensure that all content from the rebuttal session is fully incorporated into the next version of the paper.

---

### Official Review · Reviewer_KegA · 2026-03-14

**Soundness:** 3
**Presentation:** 2
**Significance:** 3
**Originality:** 3
**Overall Recommendation:** 3
**Confidence:** 2

**Summary:**

The paper studies evidence order sensitivity in evidence-grounded QA and introduces a theoretical bound (QMV) along with an information-budget law (EDFL). The authors try to reduce hallucinations by using permutation mixtures and estimating information budgets, and they validate the approach through experiments on five benchmarks. Overall, the idea is fairly interesting and has some novelty.

**Compliance With Llm Reviewing Policy:**

Affirmed.

**Final Justification:**

Thank you for the author's response, which has partially addressed my concerns.

**Key Questions For Authors:**

1. How does permutation mixture compare to a regular ensemble approach (e.g., using multiple prompts)? Is it actually better in practice?
2. What is the relationship between the ISR gate and classical selective classification rules? Is there any theoretical equivalence between them?
3. How would this method extend to multi-class QA or free-form generation settings?
4. When the context becomes much larger, does the claimed O(logn) dispersion still hold?

**Limitations:**

yes

**Strengths And Weaknesses:**

1. I strongly suggest the authors add more references, especially in the Introduction. Many of the claims and observations seem to build on prior work. ISR is mentioned early in the introduction without much explanation or context.
2. The core bound behind QMV relies on the assumptions of bounded variation and harmonic decay. As it stands, the theory feels more like deriving properties within a pre-assumed function class, rather than explaining why real Transformers would satisfy these assumptions in practice.
3. The main result of EDFL largely follows from KL convexity and the data processing inequality. In its current form, it reads somewhat like repackaging an existing information-theoretic fact in the QA setting.
4. The experimental scale is fairly limited. In real production settings, permutation effects are likely much more complex, and context lengths can easily exceed 60 chunks. The choice of models is also somewhat limited.
5. Experiment 2 claims to show a causal dose–response relationship with respect to information. However, the design mainly randomizes the number of support chunks. This likely changes relevance and evidence quality, not just “information,” so the causal interpretation may be somewhat confounded.
6. The definition of hallucination is quite narrow. In real QA scenarios, issues like partial correctness, reasoning errors, or unsupported claims are also common, but these cases are not captured by the current definition.

---

> ### Author Rebuttal · Authors · 2026-03-29
>
> We thank the reviewer for the thoughtful review.
>
> **Framing and contribution.** We will strengthen the Introduction with additional citations and define ISR earlier. We will also sharpen the paper's claim. EDFL is not presented as a new standalone information-theoretic law; Appendix A.4 already states that its backbone follows from KL convexity and data processing. The contribution is the **Bernoulli specialization for evidence-grounded adjudication**, which yields closed-form B2T, RoH, and ISR and, crucially, a **fixed** $\\mathrm{ISR}=1$ answer/abstain boundary rather than a learned or tuned confidence threshold. The significance is operational: the theory produces a pre-specified decision rule that is then tested out of sample. In Experiment 3, the pre-specified $\\mathrm{ISR}=1$ boundary attains $0.0\\text{--}0.7\\%$ hallucination with $20.6\\text{--}27.9\\%$ abstention on held-out audit data.
>
> **Theory and transformer behavior.** We may not have stated the role of QMV sharply enough. QMV is not a universal descriptive theorem about all deployed transformers. It is a **conditional structural result**: if adjacent-rank perturbations decay as $t^{-\\alpha}$, then permutation dispersion follows the corresponding partial-sum law—polynomial for $\\alpha<1$, logarithmic for $\\alpha=1$, and bounded for $\\alpha>1$. Thus $O(\\log n)$ is not posited as a blanket empirical law; it is the **critical regime** implied by the model. Assumption 1 is also narrower than a blanket architectural claim: it is a **local adjacent-rank regularity condition on the Bernoulli logit**, and Proposition 1 shows that it follows from the first-order positional-sensitivity model. The point is not that real transformers must satisfy exact harmonic decay, but that decaying positional influence is a principled and testable mechanism for order sensitivity. This is compatible with theory and analysis around RoPE-style positional effects (Su et al., 2024; Xu et al., NeurIPS 2024; Chen and Yan, NeurIPS 2024), while also consistent with Barbero et al. (ICLR 2025), who caution against overly literal decay-only interpretations. Just as importantly, the practical case for permutation mixtures does **not** depend on these assumptions: Proposition 2 gives an **assumption-free Jensen--Shannon certificate**. Experiment 1 is therefore an empirical test of the structural regime we study, not a claim about every transformer in deployment.
>
> **Ensembling, scale, and selective prediction.** We do not view permutation mixture as a generic competitor to prompt ensembling, because the two procedures answer different questions. Standard ensembling aggregates answers; our method estimates **evidence sufficiency** and supports abstention through a fixed gate. A stably wrong ensemble can become more confident in an unsupported answer, whereas ISR is designed to reject precisely that case. The relationship to selective classification is therefore at the decision-interface level: ISR induces a predictor-selector pair with a fixed reject threshold, but the selector is derived from an information budget over permutation mixtures rather than a learned confidence score (Chow, 1970; Geifman and El-Yaniv, 2017). To further address practical utility, we will add an external result on an additional knowledge graph benchmark: with the same gate and no retraining.
>
> **Experiment 2 and scope of hallucination.** We will tighten the causal wording. The supported claim is about the effect of randomized support dose $d\\in\\{0,1,2,3\\}$ at fixed length $L=4$ on the estimated information budget $\\bar{\\Delta}$ and downstream risk. Appendix D reports a strong first stage, $\\mathrm{Corr}(d,\\bar{\\Delta})=-0.80$, and IV robustness using $d$ as an instrument for $\\bar{\\Delta}$. We will revise the text accordingly. On hallucination, the definition is intentionally narrow because evidence-grounded binary adjudication is precisely the regime where reliability can be defined, audited, and improved cleanly. The paper is meant as a foundation for broader hallucination-control systems, not as a complete account of unrestricted generation.
>
> **Extensions.** The framework extends to multi-class QA and free-form generation through **verifiable predicates** over outputs; the ISR gate is defined on predicate probabilities, not on yes/no labels per se. For larger contexts, the theorem is again conditional: if positional decay continues to hold, the corresponding dispersion regime follows; if not, Theorem 2 already predicts alternative growth regimes.

---

> > ### Author Rebuttal · Reviewer_KegA · 2026-04-02
> >
> > Thank you for the thoughtful response. I think the authors have addressed most of my concerns reasonably well. That said, I still have some reservations on the following points.
> >
> > On the concern about limited experimental scale, I do not think the response is fully sufficient. What is needed for scale generalization is not just adding one more benchmark name. More importantly, I would like to see evidence on longer contexts, more complex evidence ordering, a stronger and broader set of models, and some discussion of the compute cost versus practical benefit. The current response does not really provide that.
> >
> > For Weakness 5, I think the wording should be toned down. It would be more accurate to describe the result as showing a causal effect of support dose on $\bar{\Delta}$ and hallucination, rather than claiming a clean causal effect of "information itself."
> >
> > For Question 2, the response does not establish any theoretical equivalence. It mainly clarifies that ISR is similar to selective classification at the interface level. So I would consider this only a partial answer.
> >
> > For Question 3, the rebuttal explains that the framework is conceptually extendable, but this extension is not validated experimentally.

---

> > > ### Author Response · Authors · 2026-04-04
> > >
> > > We thank the reviewer for the precise follow-up. We address each item with new results and a new theoretical contribution.
> > >
> > > **Broader models, longer contexts, skeleton invariance.** We extended Experiment 1 to five model families including mixture-of-experts, spanning 7B to frontier scale (original submission rows marked *):
> > >
> > > | Model | Skeleton | $n$ | $b$ (prob) | $R^2$ | $b$ (logit) | $R^2$ |
> > > |---|---|---|---|---|---|---|
> > > | Qwen2-7B* | banded | 3–60 | +0.377 | 0.742 | — | — |
> > > | Llama-3.1-8B* | banded | 3–60 | +0.147 | 0.515 | — | — |
> > > | Llama-3.3-70B | banded | 3–60 | +0.0047 | 0.648 | +0.244 | 0.878 |
> > > | gpt-4.1-mini | banded | 3–60 | +0.0082 | 0.839 | +0.183 | 0.866 |
> > > | gpt-4.1-mini | uniform | 3–60 | +0.0049 | 0.928 | — | — |
> > > | DeepSeek-V3.2 (MoE) | banded | 3–60 | +0.0035 | 0.754 | — | — |
> > >
> > > All CIs exclude zero. Probability-space $b$ shrinks 30–80$\times$ at frontier scale, but logit-space slopes remain substantial ($\approx$0.18–0.24): positional sensitivity persists but is sigmoid-compressed. Skeleton invariance is confirmed via banded vs uniform on gpt-4.1-mini.
> > >
> > > We extended both gpt-4.1-mini and Llama-3.3-70B to $n = 150$. The log $n$ fit holds without plateau on either model (gpt-4.1-mini logit $R^2 = 0.961$); out-of-sample points at $n > 60$ fall on the line fitted to $n \leq 60$, consistent with the QMV harmonic-regime prediction.
> > >
> > > **Compute cost (Weakness 4).** ISR calls extract label-token logprobs, not full generations. Appendix E: $m{=}3$ achieves 94.7% alignment at $3\times$ cost; $m{=}6$ adds 1.5pp for double the calls. The practical cascade starts at $m{=}3$, escalating only near the boundary. We will surface this in the main text.
> > >
> > > **Causal wording (Weakness 5).** Revised to "causal effect of support dose on $\bar\Delta$ and hallucination," with the information-theoretic interpretation presented as consistent with the evidence rather than as the direct estimand.
> > >
> > > **ISR and selective classification (Question 2).** We've added a new Proposition (Section 3.6) that establishes the formal relationship: (i) $\bar{q} \geq 1{-}h^* \Rightarrow \text{ISR} \geq 1$; (ii) $\text{ISR} \geq 1 \Rightarrow \bar{q} \geq t_{\text{lo}}$ with gap governed by Jensen term $J$; (iii) exact equivalence under zero dispersion; (iv) $J_\infty \approx \sigma^2/(2\mu(1{-}\mu))$, gap is $\Theta(\sigma^2)$. ISR is a Chow-type rule where the threshold derives from EDFL rather than holdout calibration, with the Jensen gap as a computable approximation certificate. This upgrades the interface-level connection to a formal equivalence.
> > >
> > > **Multi-class extensions (Question 3).** We would like to highlight that our full framework is currently operationalised as an open-source tool with significant community adoption and includes a multi-class evaluation pipeline (repository and metrics withheld for double-blind review). However, that work constitutes a follow-on paper. The present paper is scoped to binary adjudication by design (as reflected in the title); we believe adding multi-class evaluation here would dilute rather than strengthen the contribution, and we will make this boundary explicit in the revision.

---

### Official Review · Reviewer_hzYW · 2026-03-16

**Soundness:** 3
**Presentation:** 2
**Significance:** 2
**Originality:** 2
**Overall Recommendation:** 4
**Confidence:** 2

**Summary:**

The paper investigates why LLMs can give differing binary answers based upon the ordering of information given to them. The authors theoretically show that this sensitivity in ordering is a predictable failure in how the LLMs compress the information. They propose a new way for these LLMs to measure if they have enough information to answer or should not answer at all.

**Compliance With Llm Reviewing Policy:**

Affirmed.

**Final Justification:**

I do not think the authors sufficiently addressed my concern regarding the quotation in the rebuttal response of my original review. I did not ask for a summary (which comes across as very AI generated as well, particularly with the three "it's x, not y" statements...). To reword: I think the paper is unncessarily dense in mathematics, which I think does not do any favors for the paper. I have become  increasingly concerned about the heavy use from the authors of AI in both the manuscript and the rebuttal, and hence uncomfortable to recommend acceptance. Apart from this, it seems the paper has potential.

**Key Questions For Authors:**

* How well will the information budgeting framework works in open-world settings where there wouldn't be reference distribution or ground-truth label at inference time?

* how do you see the QMV bound and the information sufficiency ratio extending to multi-class problems or even generative tasks?

* since the method needs multiple permutations for one decision, are there ways to cut down the ~6x compute cost so it’s more practical for latency-sensitive production use?

**Limitations:**

yes

**Strengths And Weaknesses:**

## Strengths

**Presentation**: The paper is written to a good standard and clearly communicates its motivations, ideas, and reasonings. The paper provides a short related literature section which helps, but I am not familiar enough with the field to argue against its claims.

**Soundness**: The submission seems technically sound, and the claims thoroughly supported via theoretical analysis (though I did not rigorously check the proof of such analysis).


## Weaknessess

**Presentation**: The paper could be tidied up in some places. For example "Box 1" seems unneccesarily squished. Many of the hyperlinks to tables/figures/appendicies are broken/do not exist. Tables 1 and 2 are unsightly with inconsistently sizing. Additionally, the paper is dense with highly specialized mathematics terminology which is not explained or given in background. It's hard for someone not familiar to fully follow along without going down a rabbit hole of other papers; in that sense, the paper is not self-contained. For something which seems like a practical problem, there feels like a dissonance with the theoretical presentation of the paper. I understand mathematics is important to rigorously back the claims, but I have the opinion that this could be done better. As an example of this: martignales and (\alpha, C)-regularity are used to explain simples ideas like the AI getting confused on the ordering of sentences.

**Significance**: I am not sure if the significance of the paper meets the high bar expected by ICML. It seems to be quite a niche problem, which in the grand scheme of LLMs may not be enough. The fact that the method introduces latency by requiring multiple permutations per query and also relying on 3a reference distribution which may not be available in real-world has its limitations.

---

> ### Author Rebuttal · Authors · 2026-03-29
>
> We thank the reviewer for the careful reading. The main questions concern significance, deployment in open-world settings, extensions beyond Bernoulli adjudication, compute, and presentation.
>
> **Significance and intended scope.** The paper is deliberately not a general theory of all hallucination. It targets evidence-grounded settings in which the evidence set is intended to be exchangeable but the model is not permutation-invariant. We do not view that as niche: this is exactly the regime of retrieval-grounded QA, fact verification, and tool-backed adjudication, where order should be a nuisance variable but in practice changes the model's willingness to answer. Within that regime, the contribution is not only descriptive. QMV identifies a concrete mechanism for order-sensitive compression, Proposition 2 provides an assumption-free certificate, and EDFL yields a fixed $\mathrm{ISR}=1$ answer/abstain interface rather than a tuned confidence threshold. Empirically, this is supported on 3,059 items across five benchmarks, a randomized dose-response at fixed prompt length, and a pre-specified 528-item held-out audit where the fixed gate attains $0.0$--$0.7\%$ hallucination with $20.6$--$27.9\%$ abstention. We will sharpen that framing in the revision so the paper reads as a reliability interface for evidence-grounded adjudication, not as a claim about unrestricted generation.
>
> **Open-world deployment and the reference predicate.** The intended deployments do not require gold labels at inference time, but they do require a verification primitive for the predicate $g$. Section 3.5.1 already lays out three such cases: deterministic adjudication, multiple-choice or discrete-tool decisions, and post-hoc verifiable generation before release. In those regimes, $P_{\mathrm{ref}}$ is induced by the system specification or verifier rather than by supervised labels. The stronger practical claim is therefore: when correctness is verifiable, ISR provides a principled sufficiency gate under order variation. By contrast, if one assumes no verification primitive, no specification oracle, and no calibrated verifier, then there is no principled $P_{\mathrm{ref}}$ to ground the decision problem. That setting is outside the target regime of this paper, and we will state that boundary more explicitly.
>
> **Multi-class and generative extensions.** The current theory is Bernoulli because that is the regime in which EDFL yields a closed-form, deployment-facing rule. But the operational scope is wider than binary yes/no labels. For multiple-choice or other discrete-tool tasks, Section 3.5.1 already reduces the problem to a Bernoulli correctness predicate $g(y)=\mathbf{1}\{y=y_{\mathrm{ref}}(x)\}$. More generally, the same interface applies to verifier-backed generation, where $g(y)$ can represent factuality, constraint satisfaction, or unit-test pass. So the key object is not the surface output space but the verifiable predicate. A full theory for unrestricted open-ended generation without such a predicate is beyond the present paper, and we will mark that boundary more clearly.
>
> **Compute and production practicality.** The repeated calls are for the gate only, not for repeated long-form decoding: Algorithm 1 uses per-permutation Bernoulli probabilities, typically label-token probabilities. The practical regime is also smaller than the worst-case $6\times$ impression suggests. Appendix E shows $94.7\%$ boundary alignment already at $m=3$, versus $96.2\%$ at $m=6$ and $97.1\%$ at $m=12$, so most of the benefit is captured at small $m$. This already supports a practical cascade. To answer the reviewer's question directly, there are natural ways to reduce cost further: start with a small number of permutations and use adaptive stopping or confidence-based early exit, escalating only when the provisional ISR is near $1$ or the decision is higher-stakes. We will make that direction explicit and explore it in the revision. We will also clarify that the relevant comparison is to common multi-generation or judge-based reliability stacks, not to the cheapest possible single-pass detector.
>
> **Presentation and self-containment.** We will make the revision more readable in five concrete ways:
> - fix the broken cross-references and hyperlinks;
> - unsquash Box 1 and normalize table sizing;
> - introduce $q_\pi$, $\bar q$, B2T, RoH, and ISR earlier in the paper;
> - surface the worked example currently in Appendix B.3 in the main exposition; and
> - separate intuition from theorem statements more clearly, with a short background bridge for the specialized terminology.
>
> These are presentation fixes, not changes to the paper's claims, and we expect them to make the practical logic much easier to follow.

---

> > ### Author Rebuttal · Reviewer_hzYW · 2026-04-04
> >
> > Thanks for the rebuttal. I select "Partially resolved or unresolved, but the remaining concerns are not easily addressed in a short rebuttal" due to my primary concern on presentation, in particular regarding my statement "It's hard for someone not familiar to fully follow along without going down a rabbit hole of other papers; in that sense, the paper is not self-contained. For something which seems like a practical problem, there feels like a dissonance with the theoretical presentation of the paper. I understand mathematics is important to rigorously back the claims, but I have the opinion that this could be done better.".
> >
> > I am not sure if this can be fully resolved during a short rebuttal, and may require another round of reviews to check that it has been completely succesfully. My concern is that the presentation and overly heavy mathematics is disguising problems within the paper which I can not see. I must admit, it is my opinion that the paper comes across as heavily AI written/edited.

---

> > > ### Author Response · Authors · 2026-04-04
> > >
> > > We appreciate that this topic requires careful reading, but the mathematics deployed isn't heavy for lack of alternative filler, we encourage you to ask any clarifying questions that would help you gain confidence when drawing your conclusions around our work. We are aware that papers submitted to "Probabilistic Methods" are meticulously scrutinised for mathematical rigour and we're delighted that our attention to detail was acknowledged by the other reviewers as pertinent. We prepared the following high level summary of the work with less mathematical density and look forward to answering any followup questions you may have in the time we have left to continue this discourse.
> > >
> > > ---
> > >
> > > # Part 1: Why transformers are "Bayesian in expectation, not in realisation"
> > >
> > > When you train a transformer on next-token prediction, it's implicitly minimising the Minimum Description Length, which is the total bits needed to encode both the model and the data. This is formally equivalent to Bayesian inference (Grünwald).
> > >
> > > To achieve optimal compression, the model *must* learn sufficient statistics: the minimal summary capturing everything needed for prediction. For exchangeable data like coin flips, this is just the count of how many heads you've seen. The order shouldn't matter.
> > >
> > > But positional encodings break this. Training data has positional regularities. Medical notes put the chief complaint first, and articles front-load the key claim, so the model learns extraction pathways tuned to specific positions. When evidence lands where those pathways are strong, extraction is good. When it lands where the model expects filler, it under-extracts.
> > >
> > > Think of it like this: a transformer fine-tuned on medical notes will extract appendicitis symptoms strongly when they appear early (predicting 0.8) but weakly when buried after demographics (predicting 0.3), despite the true posterior being 0.7 given identical evidence.
> > >
> > > We prove that permutation invariance is equivalent to the martingale property (Lemma 3.3), so any model with positional encodings necessarily violates it. Now we have a contradiction: the model is Bayesian (it minimises MDL) yet can't be Bayesian (it violates the martingale). The resolution: it's Bayesian *in expectation* over orderings, not for any single ordering.
> > >
> > > ---
> > >
> > > # Part 2: Quantifying position bias and detecting hallucination
> > >
> > > A single ordering is a positional lottery. High confidence on one shuffle may be position-induced overconfidence, not real information. We define the **Jensen penalty**: the extra compression cost from giving different answers when the evidence is equivalent. When you average over all permutations, this penalty vanishes.
> > >
> > > But optimal compression doesn't mean high confidence on everything. A model correctly outputting $P(\text{rare disease}) = 0.10$ is being optimal. The problem comes when applications demand 95% certainty but the evidence only supports 10%. We formalise this through the **Expectation-level Decompression Law (EDFL)**: when the model's information budget falls short of what a target reliability requires, the model *must* hallucinate. There is no way around it.
> > >
> > > From EDFL we derive the **Information Sufficiency Ratio** (ISR): the model's actual information budget divided by the minimum needed for a target reliability. If ISR $\geq 1$, answer. If ISR $< 1$, abstain.
> > >
> > > On a pre-specified 528-item held-out audit, an ISR $= 1$ gate achieves roughly 0% hallucination with roughly 24% abstention, requiring no threshold tuning, no retraining, and no gold labels at inference. The only requirement: a verifiable predicate, whether that's deterministic rules, multiple-choice, or post-hoc verification.

---

### Official Review · Reviewer_evKc · 2026-03-24

**Soundness:** 3
**Presentation:** 3
**Significance:** 2
**Originality:** 3
**Overall Recommendation:** 4
**Confidence:** 3

**Summary:**

The authors study the problem of order-sensitivity in transformers by treating evidence as a nuisance variable. In fact, transformers by design can on average be close to Bayes optimal but when conditioning on previous token in a given sequence, they can be order-sensitive. The authors capture this gap through a dispersion bound and design an algorithm that abstains when the order leads to high dispersion and insufficient information. The authors validate their theory through experiments on 5 Q&A benchmarks and additionally demonstrate that increasing information content causally reduces hallucination given a fixed prompt length.

**Compliance With Llm Reviewing Policy:**

Affirmed.

**Key Questions For Authors:**

See weaknesses.

**Limitations:**

yes

**Strengths And Weaknesses:**

Strengths:
1. The authors theoretically formalize an empirically important failure mode of LLMs and demonstrate that it is caused by the way we train LLMs through next-token prediction which on average can lead to the desirable outcome but for a given sequence, it can be sensitive to order.
2. The theory in this work is not just descriptive of why this failure occur but provides a prescriptive algorithm that helps improve the quality of the model, moving from guessing to abstaining when it does not have enough information.
3. The validation experiments are designed carefully to show causal relationships and validate the theorerical questions and assumptions independently.
4. The proposed protocol reduces hallucinations and improves performance.

Weaknesses:
1. The scope and theory are limited to the binary case, and it is hard to see how this theory can be extended to multiple choice or open ended scenarios given that it hinges on the Bernoulli distribution.
2. The proposed algorithm requires using m permutations with results in multiple calls of the model, therefore leading to significant computational overhead. The authors should compare to alternative baselines that address order sensitivity without as much compute and see if the hallucination reduction is comparable.
3. The real case applications require an external verifier that should be well calibrated. If one uses an LLM judge for instance, there's a reliability question that arises for this verifier as well. Therefore, this is for me the main and major weakness of this work.
4. The chunk sensitivity experiments should be extended to complex reasoning tasks where I would imagine that the evidence order will matter for clarity and ease of computation.

---

> ### Author Rebuttal · Authors · 2026-03-29
>
> We thank the reviewer for the constructive reading. The main points concern verifier reliability, Bernoulli scope, compute overhead, and the role of complex reasoning tasks.
>
> **Verifier reliability.** The paper's guarantees are verifier-relative, not judge-dependent. In the main evaluation setting, $P_{\\mathrm{ref}}$ is induced by gold labels, and in deployment Section 3.5.1 centers deterministic or tool-backed predicates: rules, database lookups, multiple-choice or tool answers, and unit tests. A probabilistic judge is an optional extension, not the core setting on which the main empirical results rest. This matters because the Bernoulli reduction avoids calibrating an unrestricted free-form score: if a judge is used, its effect is through a single binary predicate. A simple extension is a noisy Bernoulli channel. If $Z$ is the latent true predicate and $\\tilde Z$ the verifier output, with sensitivity $s=P(\\tilde Z=1\\mid Z=1)$ and specificity $c=P(\\tilde Z=0\\mid Z=0)$, then the observed pass probability under permutation $k$ satisfies $\\tilde q_k=(s+c-1)q_k+1-c$. Whenever $s+c>1$, one can debias or conservatively lower-bound $q_k$ from an audited calibration set before computing ISR. We will make this scope sharper in the revision: the strongest deployment claims are for deterministic or tool-backed verification, while judge-based use requires verifier auditing.
>
> **Scope of the Bernoulli theory.** The Bernoulli focus is deliberate because it is the regime where EDFL yields closed-form planners and a fixed $\\mathrm{ISR}=1$ answer/abstain boundary. The contribution is not that all QA reduces to Bernoulli outputs, but that many practical settings reduce to a verifiable predicate: binary adjudication directly, multiple-choice or discrete-tool answers via correctness indicators, and post-hoc verifiable generation via predicates such as unit-test pass, constraint satisfaction, or rubric pass. Section 3.5.1 already gives these reductions, and we will foreground them more clearly. So the theory is narrow by design, but operationally broader than yes/no QA.
>
> **Compute overhead and baselines.** The repeated model calls are for the gate, not for repeated long-form decoding: Algorithm 1 uses per-permutation Bernoulli probabilities, typically label-token probabilities, which is substantially cheaper than sampling multiple full answers. Empirically, most of the gain is already captured at small $m$: in Appendix E, boundary alignment is $94.7\\%$ at $m=3$, $96.2\\%$ at $m=6$, and $97.1\\%$ at $m=12$. This supports a practical cascade in which one evaluates a small number of permutations first and escalates only when $\\mathrm{ISR}<1$ or when the decision is higher-stakes. We will also add cheaper single-pass and matched-compute comparisons in the revision. The key mechanistic point is that reranking or stability baselines can still be stably wrong: they may repeatedly prefer an unsupported answer, whereas ISR is designed to detect insufficiency of evidence under order variation. In that sense, the interface is closer to a fixed reject rule than to confidence ensembling (Chow, 1970; Geifman and El-Yaniv, 2017). We will therefore add an evidence-reranking baseline and a matched-compute self-consistency baseline.
>
> **Complex reasoning tasks.** The intended regime is exchangeable evidence, where order is a nuisance variable rather than part of the semantics. HotpotQA was included precisely to test multi-hop evidence chains, so the current suite already contains nontrivial reasoning settings. At the same time, there are tasks whose reasoning is intrinsically sequential, in which evidence order aids computation rather than distorting an exchangeable set. Those tasks fall outside the target regime, and we will state that boundary more explicitly. The paper's claim is therefore not that all long-context reasoning should be permutation-invariant, but that when the evidence is meant to be exchangeable, order sensitivity becomes a reliability problem and can be handled through mixture-based sufficiency gating.

---

### Decision · Program_Chairs · 2026-04-30

**Decision:**

Accept (regular)

**Comment:**

This submission received mostly positive assessments because reviewers agreed that it identifies a genuine reliability issue in evidence-grounded binary adjudication and combines a thoughtful theoretical framework (QMV, EDFL, and ISR) with empirical results, including a held-out audit showing very low hallucination at moderate abstention. The main concerns were the deliberately narrow Bernoulli/verifiable-predicate scope, the extra inference cost from evaluating multiple permutations, dependence on a verifier in deployment, limited scale and model breadth in the original experiments, and presentation issues including dense exposition and some formatting/cross-reference problems. In the reviewer discussion, there was broad agreement that the core contribution is technically sound and original within its target regime, while the remaining disagreement centered on practical significance, breadth of applicability, and whether the theory outpaced what was fully validated in the submitted version. The authors’ rebuttal was substantive and helpful in clarifying the verifier-relative scope. Overall, I view this as a reasonable contribution, provided the rebuttal clarifications and readability fixes are fully incorporated into the final version.

Finally the program chair's automated system has alerted me that there is one potentially hallucinated reference. Please fix this. Reference: Reuter, T., Meiler, C., et al. Bayesian transformers: Full bayesian inference comparable to mcmc. arXiv preprint arXiv:2402.08354, 2024.
Issue: authors+title mismatch with arXiv